# HARDMath: A Benchmark Dataset for Challenging Problems in Applied Mathematics

**Jingxuan Fan,**[*] **Sarah Martinson,**[*] **Erik Y. Wang,**[*] **Kaylie Hausknecht,**[*]
Jonah Brenner, Danxian Liu, Nianli Peng, Corey Wang, Michael P. Brenner
School of Engineering and Applied Sciences, Harvard University

## Abstract

Advanced applied mathematics problems are underrepresented in existing Large Language Model (LLM) benchmark datasets. To address this, we introduce **HARDMath**, a dataset inspired by a graduate course on asymptotic methods, featuring challenging applied mathematics problems that require analytical approximation techniques. These problems demand a combination of mathematical reasoning, computational tools, and subjective judgment, making them difficult for LLMs. Our framework auto-generates a large number of problems with solutions validated against numerical ground truths. We evaluate both open- and closed-source LLMs on **HARDMath-mini**, a sub-sampled test set of 366 problems, as well as on 40 word problems formulated in applied science contexts. Even leading closed-source models like GPT-4 achieve only 43.8% overall accuracy with few-shot Chain-of-Thought prompting, and all models demonstrate significantly lower performance compared to results on existing mathematics benchmark datasets. We additionally conduct a detailed error analysis to gain insights into the failure cases of LLMs. These results demonstrate limitations of current LLM performance on advanced graduate-level applied math problems and underscore the importance of datasets like **HARDMath** to advance mathematical abilities of LLMs.

## 1 Introduction

Many mathematical equations that arise in practical scientific and engineering problems cannot be solved analytically. Traditional mathematics courses tend to focus on equations with exact, analytical solutions, teaching only a limited set of techniques for solving them. Similarly, the mathematical reasoning datasets used to benchmark large language models (LLMs) are predominantly restricted to problems of this nature. However, many real-world mathematics problems involve integrals, ordinary differential equations (ODEs), and partial differential equations (PDEs) that do not have closed-form solutions and must be approached with a different set of techniques. While numerical solutions offer valuable insights, they often fail to provide intuition behind solutions behavior. A key approach in applied mathematics involves finding *approximate* analytical solutions to complex problems using asymptotic and applied analysis techniques—methods that are largely underrepresented in existing LLM benchmark datasets. To address this gap, we introduce **HARDMath**, a dataset specifically designed to focus on asymptotic reasoning in mathematics. This dataset captures a fundamentally different type of mathematical reasoning compared to other benchmarks and can be useful for evaluating LLMs' abilities to make research-relevant approximations.

**HARDMath** consists of 1,466 problems inspired by a graduate-level course on asymptotic methods. These problems cover algebraic equations, ODEs, and integrals commonly encountered in real-world scientific and engineering contexts, where exact solutions often do not exist. Instead, various asymptotic reasoning techniques are employed to find approximate but accurate solutions to these complex mathematical problems.

A primary motivation for developing **HARDMath** is the lack of benchmark datasets targeting the mathematical approximation methods required in many applications. While some recent works

---

[*]Equal contribution
Dataset: https://github.com/sarahmart/HARDMath

have begun to include university-level problems (Liu et al., 2024), most datasets focus on grade school- to high school-level mathematics problems (Amini et al., 2019; Hendrycks et al., 2021; Cobbe et al., 2021) whose solution methods only involve direct, 'clean' calculations. In contrast, **HARDMATH** targets applied mathematics problems that require *approximate* analytical solutions; an equally important yet underrepresented aspect of mathematical reasoning. Solving such problems can be challenging even for individuals with highmathematical proficiency, as it requires advanced techniques from calculus, differential equations, and complex analysis. Additionally, computational tools are often needed to analyze the behavior of different terms in each equation and to derive numerical solutions that can serve as benchmarks for the approximations. Given the difficulty of these problems and their prevalence across science and engineering—where researchers may not always have a strong background in advanced mathematics—this level of difficulty and style of mathematics should be included in LLM evaluations.

Rather than relying on the typical approach of collecting problems from textbooks, standardized tests, or competitions, as seen in most existing datasets, we developed algorithms to automatically generate problems and their step-by-step solutions. We implemented a comprehensive testing methodology to evaluate the mathematical reasoning abilities of leading LLMs in the domain of approximation methods. Our dataset includes a larger set **HARDMATH** that can be used for model developments (e.g. novel prompting techniques or fine-tuning), as well as two test sets - **HARDMATH-MINI** and **WORD-PROBLEMS-HARDMATH**, which are used to assess LLM performance. We present an evaluation accuracy summary and error mode analyses. Our results demonstrate that the performance of current LLMs on these problems is poor, highlighting significant room for their improvement on these challenging asymptotics problems.

## 2 RELATED WORK

### 2.1 MATHEMATICAL DATASETS

LLMs have shown promising capabilities in mathematics. However, evaluating and expanding the full extent of these abilities requires diverse datasets with problems that go beyond basic arithmetic or elementary word problems. Existing benchmarks often focus on these simpler domains, with a gap in addressing graduate-level applied mathematics problems that demand a deeper understanding and diverse, multi-modal analytical skills. Most mathematics datasets for evaluating or training LLMs contain samples that either present the problem directly or within a constructed narrative context. Notable examples of these datasets include **MATH** (12,500 high school competition-style problems) (Hendrycks et al., 2021), **GSM8K** (8,500 multistep grade-school problems) (Cobbe et al., 2021), **MATHQA** (37,000 GRE/GMAT-level multiple-choice problems) (Amini et al., 2019), and **ODYSSEY-MATH** (387 hand-curated problems across various difficulty levels) (Netmind.AI, 2024). While these existing datasets are valuable for assessing LLM math performance in certain areas, most are limited in scope and complexity.

Recent efforts target more advanced problems that are most often manually-sourced. Relevant works include **JEEBENCH** (Arora et al., 2023) and a subset of the **MATHBENCH** dataset (Liu et al., 2024), both of which cover some college-level topics including simple ODEs and multivariable calculus. More advanced-level problems are presented in **GHOSTS**, which contains a **GRAD-TEXT** subset—a collection of 130 exercises from graduate-level mathematics textbooks in functional analysis, topology, and probability theory (Frieder et al., 2024)—and in **ARB**, which features a small set of university-level formal mathematics problems from prior qualifying examinations in the mathematics departments at Harvard University and the University of California, Berkeley (Sawada et al., 2023). However, these datasets are limited by their size and scalability; datasets created by scraping textbooks or similar resources are generally quite small and difficult to broaden easily. Most of these challenging datasets also focus on abstract, formal mathematics and exclude other forms of mathematical reasoning. Finally, textbook problems are often protected by copyright, which can complicate their public use.

Existing datasets (summarized in Table 1) thus lack the scale and specific focus needed to evaluate LLMs on advanced mathematical problems that may be highly useful for scientific research. **HARDMATH** aims to address these limitations by offering a large collection of challenging applied mathematics problems inspired by a graduate-level course on asymptotic methods. It emphasizes problems that require diverse mathematical approaches, numerical calculations, and subjective judg-

Table 1: Comparison of **HARDMATH** with related datasets. Note that for all the datasets excluding **MATH** and **GSM8K**, we report the number of relevant problems at a comparable difficulty to our dataset (e.g., **THEORY-KNOWLEDGE-COLLEGE** in **MATHBENCH**, and **GRAD-TEXT** and **HOLES-IN-PROOFS** from **GHOSTS**.) **HARDMATH** is the largest graduate-level dataset.

| Dataset | Size | Problem Sourcing | Difficulty |
|---|---|---|---|
| **MATH** (Hendrycks et al., 2021) | 12.5K | Manual | High School |
| **GSM8K** (Cobbe et al., 2021) | 8.5K | Manual | Grade School |
| **MATHBENCH-T** (Liu et al., 2024) | 632 | Manual, Algorithmic | Undergraduate |
| **JEEBENCH** (Arora et al., 2023) | 236 | Manual | High School |
| **GHOSTS** (Frieder et al., 2024) | 190 | Manual | Graduate |
| **ARB** (Sawada et al., 2023) | 34 | Manual | Graduate |
| **HARDMATH** (Ours) | 1.4K | Algorithmic | Graduate |

ment, mirroring the complexity of problems faced by researchers in a variety of domains. Code for auto-generating the problems in **HARDMATH** can be used to generate any number of additional problems, which is a unique and powerful feature for scaling LLM benchmarking and model developments like novel prompting techniques or fine-tuning. A key area of interest in current LLM research is developing models that can effectively use external tools. The problems in our dataset are unique because they involve approximate solutions that cannot be formalized using tools like Lean or similar software. To excel in this benchmark, LLMs must integrate tool use with sophisticated reasoning. This makes **HARDMATH** particularly valuable for benchmarking and developing LLMs capable of effective tool use, setting it apart from other mathematical datasets.

## 2.2 RECENT INTEREST IN ADVANCED MATHEMATICAL REASONING

As LLMs continue to improve, there has been growing interest in developing more challenging benchmarks, especially in mathematics. A notable example is the recent open challenge, *Humanity's Last Exam*, which aims to create the world's most difficult public AI benchmark, requesting questions that "only exceptional individuals can answer correctly," do not involve "straightforward calculation/computation," and are written by individuals with PhD-level academic training (Hendrycks & Wang, 2024). Similarly, frontier models have been advancing quickly, and many are explicitly focused on quantitative and scientific reasoning, such as OpenAI's recent o1 series. In line with our motivation for developing **HARDMATH** to better track the progress of LLMs, OpenAI argues that "recent frontier models do so well on **MATH** and **GSM8K** that these benchmarks are no longer effective at differentiating models" (OpenAI, 2024).

## 3 DATASETS

### 3.1 HARDMATH DESIGN CHOICES

Here, we detail the **HARDMATH** dataset, which contains problems on polynomial nondimensionalization, polynomial root-finding, ODEs, integrals, and word problems that contextualize each of these. A sample of a **HARDMATH** integral problem is shown in Box 1. **HARDMATH** contains four problem classes with seven distinct problem types, as well as 40 handwritten word problems contextualizing the problem types. The main **HARDMATH** dataset, which can be used for model developments like fine-tuning, contains 1,060 problems, and the evaluation dataset **HARDMATH-MINI**, which we use in this paper to benchmark LLM performance, contains 366 problems. Figure 1 provides a detailed breakdown of these datasets.

---

**Box 1: Sample Integral Problem and Solution**

**Problem:**
Consider the integral $I(\epsilon) = \int_0^{56.00} \frac{1}{\epsilon + 2.0x^{6.0} + 2.0x^{9.0} + 5.0x^{11.0} + 5.0x^{13.0}} dx$. Develop analytical formulas that approximate $I(\epsilon)$ for different regimes of $\epsilon$.

---

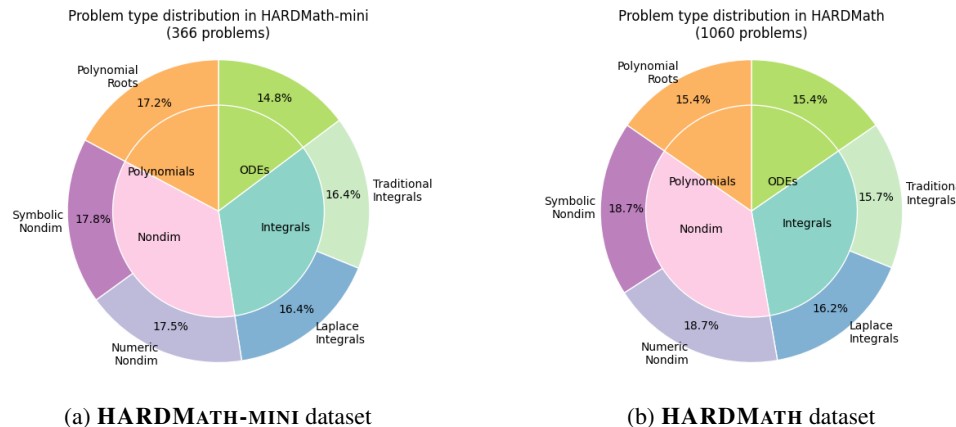

(a) **HARDMATH-MINI** dataset      (b) **HARDMATH** dataset

Figure 1: Breakdowns of the **HARDMATH-MINI** (left) and the **HARDMATH** (right) datasets.

**Solution:** The integral is of the form $I(\epsilon) = \int_0^{56} \frac{1}{\epsilon + P(x)} dx$ where $P(x)$ is a polynomial. Thus, **its value can be estimated as the product between a height and a width**.

Since the integrand is maximized at $x = 0$, the height can be set to $\frac{1}{\epsilon}$.

For **small $\epsilon$**, we define the width as the point where the integrand becomes half of its maximum height. This corresponds to solving for $x$ given $P(x) = \epsilon$. Applying **dominant balance**, considering the term in $P(x)$ with the smallest degree, the width is approximated as $\left(\frac{1}{2.0*\epsilon}\right)^{1/6.0}$. Therefore, the analytical approximation of the integral for small $\epsilon$ is $I(\epsilon) = \frac{0.8909}{\epsilon^{0.8333}}$.

For an **intermediate regime** where $\epsilon$ is large, we also define the width based on the term with the largest degree. The width is approximated as $\left(\frac{1}{5.0*\epsilon}\right)^{1/13.0}$. Therefore, the analytical approximation of the integral for large $\epsilon$ is $I(\epsilon) = \frac{0.7647}{\epsilon^{0.8333}}$.

If the width of the integral exceeds the range of integration, we consider one more regime for **very large $\epsilon$**. The width is then just the range of integration, so in this regime, the integral can be approximated as $\frac{L}{\epsilon}$. Therefore, the analytical approximation of the integral for very large $\epsilon$ is $I(\epsilon) = \frac{56}{\epsilon}$.

Altogether, the solutions at small, large, and very large $\epsilon$ are $\boxed{\frac{0.89}{\epsilon^{0.83}}, \frac{0.76}{\epsilon^{0.83}}, \frac{56}{\epsilon}}$.

One key commonality between all **HARDMATH** problems is the use of the *Method of Dominant Balance* in calculating solutions. This reduces an equation to only the terms that 'dominate' the behavior of the solution and can significantly simplify the equation (Bender & Orszag, 2013). In addition to the Method of Dominant Balance, our problems also involve other sophisticated mathematical techniques, such as checks for self-consistency and the use of numerical methods. The combination of these tools captures several key aspects of mathematical modeling, including the combined use of computational and analytical techniques. Additionally, subjective choices about the regimes of solution space to consider, the number of terms to include in approximate expressions, and the approximation methods themselves must be made on a case-by-case basis with rigorous mathematical justification. Both of these aspects are potentially difficult tasks for existing LLMs. In Box 1, we highlight the methods and regimes relevant to solving a sample integral problem.

### 3.2 DATASET GENERATION AND VERIFICATION

The dataset generation procedure is outlined in Fig. 2. Code for data generation uses `SymPy` (Meurer et al., 2017), a library for symbolic mathematics, and `SciPy`, a library for scientific computing (Virtanen et al., 2020), to implement the mathematical procedures required for obtaining approximate, analytical solutions. Problems are generated by combining randomly selected coefficients, functional forms, and initial conditions uniquely defined for each problem (described in Appendix A)—no duplicate problems are included. Solutions are generated by navigating through a set of possible cases during the algorithmic problem-solving strategy. Each mathematical step is embedded in explanatory text so **HARDMATH** solutions match the style and rigor of traditional problem set solutions. The main results for all problems are included in boxed environments in the solution explanations to

distinguish them from the rest of the text. This follows the formatting convention used in other mathematics datasets designed for LLM benchmarking, such as **MATH** (Hendrycks et al., 2021).

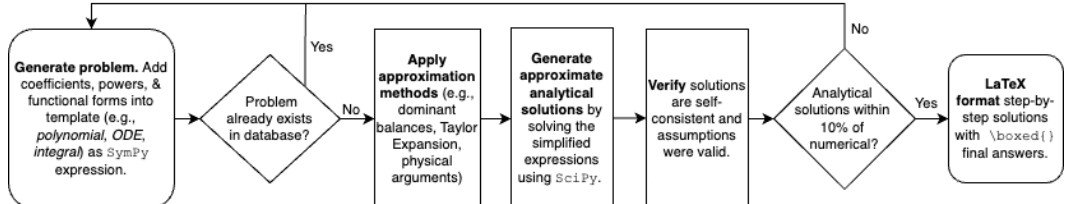

Figure 2: Flowchart detailing the data generation procedure for **HARDMATH** problems.

For each problem type, the dataset includes: 1) LATEX-formatted problem statements with prompts, 2) LATEX-formatted solution steps and final analytical answer(s), 3) demonstration of the accuracy of the analytical results by comparing with numerical solutions, and 4) metadata descriptors of the problem and solution types. For every problem type, we select evaluation points in each solution regime and calculate the relative error between the analytical solution and the numerical solution at these points. Problems were included in **HARDMATH** only if their approximate solutions had less than 10% error from the numerically calculated ground-truths. For the polynomial root correction problems, we also confirm that the corrections improve the original approximation. While manually verifying each solution step-by-step is impractical for a dataset of this size, our validation process ensures a high level of confidence in the accuracy of the solutions provided. Verification of **HARDMATH-MINI** is performed by plotting analytical solutions against numerical ground truths for a range of values in each regime (see example in Appendix A.1). This semi-automated step provides an easy means for visual human-verification—plots will directly reveal whether analytical and numerical solutions correspond in the correct solution regimes.

### 3.3 PROBLEM TYPES

For all problem types discussed in this section, details regarding parameters used to generate problems and mathematical solution techniques are provided in Appendix A.2.

#### 3.3.1 NONDIMENSIONALIZATION OF POLYNOMIALS

Nondimensionalization is a technique to simplify equations by reducing the number of parameters (Evans, 1972). In **HARDMATH**, the first type of polynomial used for nondimensionalization demonstration contains symbolic coefficients and is of the form

$$a_1 x^{n_1} + a_2 x^{n_2} + a_3, \ \ n_1 > n_2 > 0. \tag{1}$$

Nondimensionalization converts this to the form $\epsilon y^{n_1} + y^{n_2} + 1$. The second type contains numerical coefficients and is of the form

$$\pm a_1 x^{n_1} \pm a_2 x^{n_2} \pm a_3, \ \ n_1 > n_2$$

which can be simplified to $\epsilon y^{n_1} \pm y^{n_2} \pm 1$ given a specific numerical value of $\epsilon$.

#### 3.3.2 POLYNOMIAL ROOT-FINDING

Exact formulas exist for quadratic, cubic, and quartic equations, but deriving them for quintic or higher-order polynomials is not possible (Stewart, 2015). **HARDMATH** includes approximate root-finding examples for higher order polynomials of the form $\epsilon x^{n_1} \pm x^{n_2} \pm 1$ (example in Appendix A.2.2). The goal is to solve for roots in terms of $\epsilon$ using the method of dominant balance for small and large positive $\epsilon$ regimes.

#### 3.3.3 POLYNOMIAL ROOT CORRECTION TERMS

The use of two-term dominant balances—such as in the previous problem type—neglects terms and introduces an error. We can calculate a correction term $\delta$ to reduce this error. Suppose the true roots

$x^*$ of a polynomial are given by $x^*(\epsilon) = \overline{x}(\epsilon) + \delta$, where $\overline{x}$ is our approximation to the root and $\delta$ is the error term. Plugging the roots $x^*(\epsilon) = \overline{x}(\epsilon) + \delta$ into the polynomial allows one to use a Taylor expansion of $\delta$ around $\overline{x}$ to solve for the correction $\delta$. Appendix A.2.3 shows a full worked solution.

### 3.3.4 NONLINEAR ORDINARY DIFFERENTIAL EQUATIONS

We generate nonlinear third-order ODEs for which there do no exist exact analytical solutions and provide approximate formulae for small and large $x$ regimes, where the small $x$ regime is near $x = 0$ and the large $x$ regime typically involves the solution diverging (example in Appendix A.2.4). The method is robust for higher-order problems, but for simplicity **HARDMATH** includes only third-order ODEs.

### 3.3.5 TRADITIONAL INTEGRALS

We consider integrals of the form $I(\epsilon) = \int_0^a \frac{1}{\epsilon + P(x)} \, dx$, where $P(x)$ is an arbitrary polynomial. **HARDMATH** provides approximations of each integral in three regimes: small, intermediate, and large $\epsilon$. A full example is in Appendix A.2.5.

### 3.3.6 LAPLACE INTEGRALS

We consider integrals of the form $I(x) = \int_a^b g(t)e^{\pm xf(t)}dt$, which can be approximated using Laplace's Method when $x$ is very large because the integral's value is dominated by the region around $t_0$ (Bender & Orszag, 2013). Depending on where the minimum is, the approximation is either

$$I(x) \approx g(t_0)e^{\pm xf(t_0)}\sqrt{\frac{2\pi}{x|f''(t_0)|}} \quad \text{or} \quad I(x) \approx \frac{g(t_0)e^{\pm xf(t_0)}}{x|f''(t_0)|}.$$

See and Appendix A.2.6 for examples of a Laplace integral problem with solutions.

### 3.4 WORD PROBLEMS IN CONTEXT

One motivation for creating **HARDMATH** is to help LLMs recognize and solve problems where approximation techniques are needed. To evaluate how LLMs perform on such problems in realistic scenarios, we develop a smaller dataset of 40 manually-generated word problems (example in Box 2). Although this dataset is smaller than our hand-verified evaluation set, it is large enough to evaluate the effect of additional context in the problem statement on LLM accuracy.

---

**Box 2. Sample Word Problem with Context**

The density of fish at different points along a certain path in a lake can be modeled as $(\epsilon + x^2 + x^5)^{-1}$, where $x$ represents the distance from the shore in kilometers (ranging from 0 to 100 km), and $\epsilon$ represents environmental factors that affect the fish density. To study the total presence of fish along the path, develop an approximate analytical formula for $I(\epsilon)$ given below:

$$I(\epsilon) = \int_0^{100} \frac{1}{\epsilon + x^2 + x^5} \, dx.$$

---

### 3.5 AUTOMATIC GENERATION OF CONTEXT FOR WORD PROBLEMS

Recognizing that manual context crafting for these problems is labor-intensive and lacks scalability, we conduct preliminary experiments on automatically generating contexts for word problems using a powerful closed-source LLM, such as GPT-4o. The generation process involves two steps: (1) We create a foundational set of mathematical problems and solutions following the methodology in Fig. 2. These problems serve as a starting point, capturing essential mathematical formulation without context. 2) We then prompt the LLM to generate real-world contexts to embed these problems and solutions based on a specified domain seed (e.g., physics). To ensure physical plausibility, a secondary verification LLM checks parameter ranges (e.g., energy must be non-negative) and assigns a 0-1 plausibility score. A score around 0.5 indicates that while no hard violations were found, the verifier cannot fully assess violations of less stringent domain priors. Only plausibity scores >0.5 are thus

considered. Example prompts for both generator and verifier are available in Appendix A.2.7, Table 3. Using *ODEs* as a demonstration, we batch generate 30 problems with `{domain_seed}` **physics**. Appendix A.2.8 shows an example of the original vs. context-embedded problem side by side, and Appendix A.2.9 6a demonstrates the set of generated problems has good context diversity, covering different sub-fields in **physics**. Moreover, most problems' plausibility scores >0.5 , indicating queried math equations make good sense under generated contexts (A.2.9 6b).

While we recognize limitations in capturing more subtle, domain-specific priors, this approach offers a promising step toward automating the generation of applied math problems in real scientific contexts. We plan to refine these methods in future work.

## 4 EVALUATION

### 4.1 EVALUATION PROTOCOLS

We conduct evaluations of various LLMs on **HARDMATH-MINI**, a carefully curated subset of 366 problems that matches the statistical composition of **HARDMATH** (Fig. 1). This smaller dataset is designed to optimize computational resources while retaining a sufficient number of questions to ensure consistent and reliable testing outcomes, thus maintaining the integrity of our evaluation. The evaluation focuses on four distinct problem types: 1) *Nondim* includes nondimensionalization in symbolic and numerical form; 2) *Roots* includes polynomial root-finding; 3) *ODEs* includes nonlinear ODEs; and 4) *Integrals* includes traditional and Laplace integrals. The input prompt for each problem contains the essential problem setup and a detailed description of the question. Additionally, hints specific to each problem type are provided to guide the format of the answer. When few-shot prompting is used, it adds a fixed set of paired problem-solution examples from the corresponding problem types. Example prompts can be found in Appendix A.3.1, Table 7.

We evaluate model-generated responses by scoring them for accuracy using a combined protocol of automatic final answer assessment and procedural LLM-based grading. The automatic assessment follows methodology from Hendrycks et al. (2021), where models are prompted to enclose their final answers using the LaTeX `\boxed{}` command (Table 7). Evaluation then compares the model's output within the `\boxed{}` command to the dataset solution. To handle different mathematical expression formats, we implement both `SymPy`-based equivalence checks and numerical evaluations.

In addition to the standard automatic assessment of final answers, we develop a novel procedural grading approach leveraging LLMs, tailored to the unique evaluation challenges of our dataset: 1) Some problem types require complex, multi-step solution procedures (e.g. determining critical point in Laplace integral approximation) where a single cut-off criterion at the final answer cannot capture the full spectrum of model performance. Thus, grading intermediate steps in the solution procedure is necessary for comprehensive assessment. 2) **HARDMATH** targets the models' ability to make human-like abstraction and approximation judgments. Some problem types allow a narrow range of solutions rather than a single exact one, as long as the reasoning is self-consistent and the final result falls within certain threshold to numerical ground truth.

Inspired by LLMs' ability to generate consistent ratings for response content and style (Hackl et al., 2023), we employ GPT-4o as a procedural grader. The model is prompted with a ground truth answer key and grading rubrics adapted from course grading guidelines for each problem type (example grading prompts in Appendix A.3.2 Table 5). We manually verify a subset of grading responses and found that LLM-based grading is closely aligned with human grading. Average score adjustment for each model and problem type is summarized in Appendix A.3.3 Table 6. We implement this procedural grading alongside automatic answer assessment for the problem types *Roots*, *ODEs*, and *Integrals*.

### 4.2 MODEL CHOICE

We compare the performance of several closed- and open-source models on **HARDMATH** in zero- and few-shot settings with the Chain-of-Thought (CoT) (Wei et al., 2023) prompting. Closed-source LLMs include GPT-3.5 (Radford et al., 2018; 2019; Ouyang et al., 2022), GPT-4 (Achiam et al., 2023) and o1-mini (OpenAI, 2024a), open-source LLMs include Llama3 (AI, 2024) and CodeLlama

(Meta, 2023). We believe this subset of models to be representative of current LLM capabilities. We provide the prompts and hyper-parameters for LLMs evaluations in Appendix A.3.4 Table 7.

## 4.3 QUANTITATIVE RESULTS

We present the accuracy of the models and prompting settings for each problem type and the combined evaluation set (Table 2, Figure 7). Few-shot CoT prompting significantly boosts performance for all models, with o1-mini and GPT-4 showing the greatest improvement, consistent with (Wei et al., 2023) (Figure 7a). Interestingly, although the o1-mini official prompting guide recommends simple prompting over CoT (OpenAI, 2024b), we observe fairly large performance increase for all problem types at 5 shot CoT compared to 0 shot. Performance increase with prompting behavior also shows problem type-specific patterns: Figure 8 demonstrates that performance saturates quickly for harder problem types like *ODEs*. The varying performance increases among different problem types may be due to different error modes in model answers, which we discuss in the following section. It's notable that o1-mini, though with much smaller parameter size, shows considerably better performance at all tested shot levels, confirming its optimized ability for STEM reasoning (OpenAI, 2024a).

Among closed-source models, o1-mini with 5-shot CoT prompting achieves the highest overall accuracy of 62.3%. GPT-4 at 5-shot CoT scores only 43.8%. Among open-source models, Llama3-8b with 5-shot CoT prompting achieves the highest overall accuracy of 20.2%. We discuss the performance of these representative models—o1-mini, GPT-4 and Llama3—on **HARDMATH-MINI** in comparison with established datasets, including **GSM-8K** (Cobbe et al., 2021), **MATH** (Hendrycks et al., 2021), and more advanced mathematics datasets like **GHOSTS** (Frieder et al., 2024).

Llama3-8b achieves a test accuracy of 30.0% on the **MATH** dataset with 4-shot CoT and 79.6% on the **GSM-8K** dataset with 8-shot CoT prompting (AI, 2024). Testing Llama3-8b on **HARDMATH-MINI** results in an overall accuracy of 20.2% with 5-shot CoT prompting. GPT-4 (gpt-4-turbo-2024-04-09) is reported to achieve 72.2% accuracy on the **MATH** dataset with 0-shot CoT prompting (OpenAI, 2024) and 92.0% on the **GSM-8K** dataset with 5-shot CoT prompting (Achiam et al., 2023). On the **MINIGHOSTS** dataset, which also covers graduate-level mathematics, GPT-4 reaches an average score of 4.15 out of 5. We test GPT-4 on our **HARDMATH-MINI** dataset and obtained an overall accuracy of 43.8% with 5-shot CoT prompting.

Finally, we include results on OpenAI's new o1-mini, which is reported to achieve 90.0% accuracy on **MATH-500** with 0-shot CoT (OpenAI, 2024a). Testing o1-mini on **HARDMATH-MINI** reveals a significant performance increase compared to results on other models on some (e.g. *Nondim*) but not all problem types. Overall accuracy with 5 shot CoT reaches 62.3%, substantially lower compared to performance on existing mathematics benchmarks. This indicates that the **HARDMATH** benchmark consists of problems that are still challenging and unfamiliar to even the most performant LLMs developed specifically for advanced reasoning.

### 4.3.1 EXTENSIONS TO WORD PROBLEMS

To assess LLM's ability to solve similar applied math problems in real science and engineering contexts, we also test GPT-4 (best performing model with a stable version) on a set of 40 hand-crafted word problems that included a mixture of *Nondim*, *Roots*, *ODEs*, and *Integrals*. We avoided additional prompt engineering, omitting the problem-specific hints listed in Table 4. This evaluation resulted in an overall accuracy of 28.1%. We plan to leverage the automated generation method as a basis to expand the number of word problems for future work.

## 4.4 FINE-GRAINED RESULTS

In addition to reporting the summarized test accuracy, we study the detailed breakdown of model responses at different correctness levels and summarize specific error modes of LLMs solving these challenging applied mathematics questions. This analysis helps us compare performance nuances and understand reasoning paths by model, prompting technique and question type.

We first break down model performance by percentage of correct, partial and incorrect responses (Figure 3). This analysis reveals how few-shot prompting enhances model performance across varying problem types but through different strategies. Overall quantitative results already show that *ODEs*

Table 2: Evaluation Accuracy (percentage) on the **HARDMATH** evaluation set.

| Model | ALL | Nondim | Roots | ODEs | Integrals |
|---|---|---|---|---|---|
| **Closed-source models** | | | | | |
| GPT-3.5 (0 shot) | 6.04 | 5.05 | 17.2 | 1.39 | 3.33 |
| GPT-3.5 (1 shot CoT) | 14.2 | 6.11 | 29.3 | 6.94 | 18.2 |
| GPT-3.5 (5 shot CoT) | 24.6 | 24.3 | 35.0 | 16.2 | 23.1 |
| GPT-4 (0 shot) | 14.0 | 6.04 | 33.7 | 7.87 | 14.9 |
| GPT-4 (1 shot CoT) | 37.6 | 36.5 | 52.8 | 15.9 | 40.5 |
| GPT-4 (5 shot CoT) | 43.8 | 48.6 | 57.3 | 21.7 | 41.4 |
| o1-mini (0 shot CoT) | 29.8 | 38.1 | 24.3 | 10.2 | 32.5 |
| o1-mini (5 shot CoT) | 62.3 | 84.5 | 62.1 | 30.6 | 46.5 |
| **Open-source models** | | | | | |
| Llama3-8b (0 shot) | 3.67 | 0.50 | 11.5 | 4.63 | 2.52 |
| Llama3-8b (5 shot CoT) | 20.2 | 17.9 | 17.1 | 12.0 | 28.1 |
| CodeLlama-13b (0 shot) | 1.94 | 0.00 | 8.73 | 1.85 | 0.50 |
| CodeLlama-13b (5 shot CoT) | 9.79 | 8.41 | 13.1 | 9.7 | 9.57 |

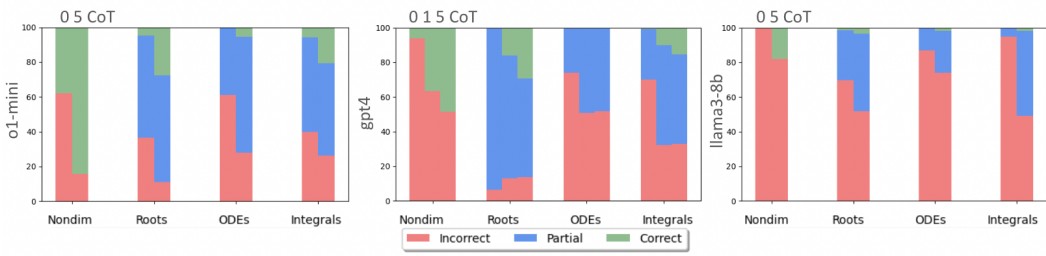

Figure 3: Percentage of correct, partial, and incorrect responses for o1-mini, GPT-4 and Llama3, prompting conditions, and problem types.

are comparatively harder for all models while *Nondim* problems appear to be the easiest (Figure 7b). For hard problems like *ODEs*, full correctness is rare. Correctness level analysis shows that models tend to increase partial credit responses with CoT prompting, as they struggle to solve the problems entirely but manage to partially address them—in this case, starting with the easier small $x$ regime solutions. In contrast, for simpler problems like *Roots*, advanced models like o1-mini and GPT-4 get more fully correct responses with increasing CoT shot number, demonstrating the models' ability to understand the approximation reasoning procedure fully (Fig. 3).

Second, we summarize the error modes of partial and incorrect responses to better understand the model's reasoning pitfalls. Specifically, we want to dissect how CoT changes model performance on the level of detailed errors. Figure 4 uses GPT-4's responses at 0 vs. 5 shots on problem type *Roots* as an analysis example. This illustrates how 5 shot CoT prompting significantly alters the error structure compared to 0 shot. The most common error mode—incorrectly setting up dominant balance by considering only the leading term—diminishes substantially. Instead, errors shift to more nuanced issues: 1) setting up correct dominant balances but missing certain cases, or 2) failing to calculate complex roots (examples of those error modes included in Box 3 and Appendix A.4.2). This shift indicates that CoT prompting improves the model's understanding and application of dominant balance techniques, enabling it to move beyond intuitive yet incorrect simplifications. We are particularly curious to compare o1-mini's error modes with existing models, given its specialization in STEM reasoning. However, in the current evaluation, o1-mini sometimes returns only the final answer without showing intermediate steps, making it difficult to trace the source of errors. We therefore leave this exploration for future work.

> ### 3. Grading Response: *Roots*
>
> **Question:** Consider the polynomial $P(x) = \epsilon x^8 + x^4 - 1$. Find approximate expressions for all roots of the polynomials in the limit of small positive $\epsilon$ and large positive $\epsilon$. Only a single term approximation to the root is required.

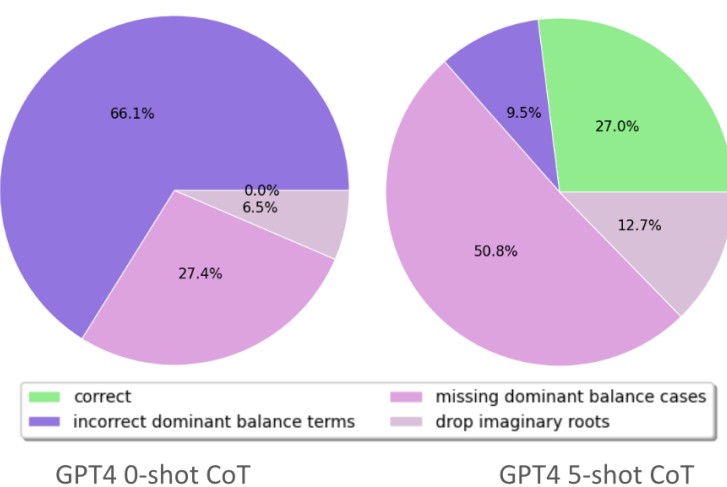

Figure 4: GPT-4 error modes for problem type *Roots* at 0 vs. 5 shot CoT prompting

---

**Grading for Small Positive $\epsilon$**:
**Model Response:** For small $\epsilon : [1, -1, i, -i]$
**Ground Truth**:

$$\text{For small positive } \epsilon : \left[ -\sqrt[4]{-\frac{1}{\epsilon}}, \; \sqrt[4]{-\frac{1}{\epsilon}}, \; -i\sqrt[4]{-\frac{1}{\epsilon}}, \; i\sqrt[4]{-\frac{1}{\epsilon}}, \; -1, \; 1, \; -i, \; i \right]$$

The response only includes the roots from the balance $B + C = 0$ and completely misses the roots from the balance $A + B = 0$. Therefore, score for small positive $\epsilon$ is $\boxed{0.5}$

---

## 5 CONCLUSION

We introduce **HARDMATH**, a new dataset covering several problem types from an advanced applied mathematics course that can be used to benchmark LLMs' mathematical capabilities and perform model developments, including fine-tuning. This dataset consists of 1060 examples, and we additionally include 366 human-verified examples in **HARDMATH-MINI** and 40 human-verified 'problems in context' that we use to evaluate various leading LLMs. **HARDMATH** is unique in several ways. First, there do not exist large-scale mathematical datasets covering problems of similar difficulty from applied mathematics. Second, **HARDMATH**'s problems and solutions are algorithmically generated, with automatic numeric validity checks and an easy visual means for human-verification, meaning that one could produce datasets of arbitrary size using our framework. This feature of **HARDMATH** is especially unique, since most existing mathematical datasets require manual problem-setting or curation from existing sources (many of which are not publicly accessible).

Our evaluation highlights that while few-shot CoT prompting significantly improves model performance, especially for models like o1-mini and GPT-4, the overall accuracy on **HARDMATH-MINI** problems remains much lower compared to other existing benchmarks. This suggests that our dataset poses unique and challenging tasks that go beyond the boundaries of current LLM capabilities, particularly in approximation-oriented mathematical reasoning. These findings emphasize the need for further improvement in LLMs to address hard math problems.

Our evaluation results use **HARDMATH-MINI** as a comprehensive test set; however, future work will fine-tune LLMs on the larger **HARDMATH** to improve performance. Additionally, while we have evaluated several frontier models, we plan to extend our evaluations to even more LLMs as they become available. This expanded evaluation should provide more detailed insights into performance disparities across different models, further advancing our understanding of LLMs' capabilities in handling complex mathematical reasoning.

ACKNOWLEDGMENTS

We thank the students who participated in the initial stages of the Fall 2023 AM 201 final project. Thanks also to the Harvard Medical School Research Computing Consultant Group for their consulting services, which facilitated the computational analyses detailed in this paper.

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

## A  APPENDIX

### A.1  SEMI-AUTOMATED SOLUTION VERIFICATION

Figure 5 provides a visual comparison of the numerical and approximate solutions for the Laplace Integral example in Appendix A.2.6 over a large domain. This allows for semi-automated human verification that analytical solutions correspond well with numerical ground truths, a method used to verify the problems in **HARDMATH-MINI**.

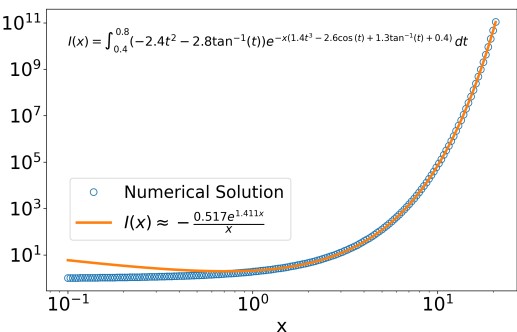

Figure 5: Visual comparison of numerical and approximate analytical solutions to a sample Laplace integral problem for solution verification.

### A.2  IMPLEMENTATION AND METHOD DETAILS FOR DATA GENERATION

The following subsections detail the process used to generate the problems and solutions for each problem type.

### A.2.1  NONDIMENSIONALIZATION OF POLYNOMIALS

The first nondimensionalization sub-type is generalized by varying the integer values for the degrees $n_1$ and $n_2$ within the range $0 < n_2 < n_1 < 10$, while keeping $a_1, a_2, a_3 > 0$ symbolic. Solutions to these problems express the dimensionless parameter $\epsilon$ in terms of these three coefficients.

---

**Sample Symbolic Nondimensionalization Problem and Full Solution**

**Problem:** Nondimensionalize the polynomial

$$a_1 x^{10} + a_2 x^9 + a_3$$

into one of the form $\epsilon y^{10} + y^9 + 1$. Express $\epsilon$ as a function of $a_1$, $a_2$, and $a_3$.

---

**Solution:** We begin with the substitution

$$x = y \sqrt[9]{\frac{a_3}{a_2}}$$

This gives the expression

$$a_1 y^{10} \left( \frac{a_3}{a_2} \right)^{\frac{10}{9}} + a_3 y^9 + a_3$$

Divide by the coefficient remaining in front of the constant, leaving us with the nondimensionalized polynomial with coefficients in terms of $a_1$, $a_2$, and $a_3$:

$$\boxed{\frac{a_1 y^{10} \left( \frac{a_3}{a_2} \right)^{\frac{10}{9}}}{a_3} + y^9 + 1.}$$

---

By inspection, we can see that

$$\epsilon = \frac{a_1 \left(\frac{a_3}{a_2}\right)^{\frac{10}{9}}}{a_3}.$$

The second subtype implements integer numerical values for the coefficients $a_1$, $a_2$, $a_3$ that are are randomly chosen from $[-10, 10]$.

---

**Sample Numeric Nondimensionalization Problem and Full Solution**

**Problem:** Nondimensionalize the polynomial

$$P(x) = 2x^7 + 8x^2 + 5$$

into a polynomial of the form $\epsilon y^7 \pm y^2 \pm 1$. Solve for $\epsilon$.

---

**Solution:** For now, we ignore the numeric values of the coefficients and instead call them $a_1, a_2, a_3$. Our polynomial is then:

$$a_1 x^7 + a_2 x^2 + a_3.$$

Use the substitution

$$x = y\sqrt{\frac{a_3}{a_2}},$$

which gives the expression

$$a_1 y^7 \left(\frac{a_3}{a_2}\right)^{\frac{7}{2}} + a_3 y^2 + a_3.$$

Divide all terms by the coefficient remaining in front of the constant term, giving us the nondimensionalized polynomial with coefficients in terms of $a_1, a_2, a_3$:

$$\frac{a_1 y^7 \left(\frac{a_3}{a_2}\right)^{\frac{7}{2}}}{a_3} + y^2 + 1$$

Substituting in the known numeric values for $a_1, a_2, a_3$ (using their absolute values as we have already accounted for sign), we get:

$$\frac{25\sqrt{10}y^7}{1024} + y^2 + 1$$

From inspection of this nondimensionalized equation, we can now identify $\epsilon$:

$$\epsilon = \frac{25\sqrt{10}}{1024} \implies \boxed{\epsilon \approx 0.08.}$$

---

### A.2.2 Polynomial root-finding

As with the nondimensionalization problems, degrees in the polynomial are randomly generated with maximum order ten and $0 < n_2 < n_1$. See a full problem and solution below.

---

**Sample Polynomial Root-finding Problem and Full Solution**

**Problem:** Consider the polynomial

$$P(x) = \epsilon x^6 - x^5 + 1.$$

Find first order approximations for all roots of the polynomials in the limit of small positive $\epsilon$ and large positive $\epsilon$.

---

**Solution:** We begin by equating the polynomial to zero to solve for the roots: $P(x) = 0$. This problem can be rewritten in the form $A + B + C = 0$, where: $A = \epsilon x^6$; $B = -x^5$; $C = 1$.

This problem has no analytical solutions, so we find approximate solutions to the roots by considering the three possible dominant balances. For each dominant balance, we find the roots of the resulting equation and evaluate whether each balance is self-consistent for small or large positive $\epsilon$.

We start with the balance $A + B = 0$, assuming that $|C|$ is negligible when compared to $|A|$ and $|B|$. Solving this for $x$ in terms of $\epsilon$ then gives us 1 non-zero root:

$$\epsilon x^6 - x^5 = 0$$

$$\implies \boxed{x = \left[\frac{1}{\epsilon}\right].}$$

To verify that these roots are consistent with the assumption that $|A|, |B| \gg |C|$, we substitute these found roots back into the terms $A$, $B$, and $C$ and compare their magnitudes. Using this method, we find that it is true that these roots are valid for small $\epsilon$, while validity for large $\epsilon$ is false.

Therefore, these roots are valid in the limit of small positive $\epsilon$ only.

Next we examine the balance $B + C = 0$, assuming that $|A|$ is negligible when compared to $|B|$ and $|C|$. Solving this for $x$ in terms of $\epsilon$ gives us 5 non-zero roots:

$$1 - x^5 = 0$$

$$\implies \boxed{\begin{aligned} x = &1, \ -\frac{1}{4} + \frac{\sqrt{5}}{4} - \frac{i\sqrt{2\sqrt{5}+10}}{4}, \ -\frac{1}{4} + \frac{\sqrt{5}}{4} + \frac{\sqrt{-10-2\sqrt{5}}}{4}, \\ &-\frac{\sqrt{5}}{4} - \frac{1}{4} - \frac{i\sqrt{10-2\sqrt{5}}}{4}, \ -\frac{\sqrt{5}}{4} - \frac{1}{4} + \frac{i\sqrt{10-2\sqrt{5}}}{4} \end{aligned}}$$

To verify that these roots are consistent with the assumption that $|B|, |C| \gg |A|$, we substitute these found roots back into $A$, $B$, and $C$ and compare their magnitudes. Using this method, we find that it is true that these roots are valid for small $\epsilon$, while validity for large $\epsilon$ is false. Therefore, these roots are valid in the limit of small positive $\epsilon$ only.

Finally, we examine the balance $A + C = 0$, assuming that $|B|$ is negligible when compared to $|A|$ and $|C|$. Solving this for $x$ in terms of $\epsilon$ gives us 6 non-zero roots:

$$\epsilon x^6 + 1 = 0$$

$$\implies \boxed{\begin{aligned} x = \Bigg[ &-\sqrt[6]{-\frac{1}{\epsilon}}, \ \sqrt[6]{-\frac{1}{\epsilon}}, \ \frac{\sqrt[6]{-\frac{1}{\epsilon}}\left(-1-\sqrt{3}i\right)}{2}, \\ &\frac{\sqrt[6]{-\frac{1}{\epsilon}}\left(-1+\sqrt{3}i\right)}{2}, \ \frac{\sqrt[6]{-\frac{1}{\epsilon}}\left(1-\sqrt{3}i\right)}{2}, \ \frac{\sqrt[6]{-\frac{1}{\epsilon}}\left(1+\sqrt{3}i\right)}{2} \Bigg] \end{aligned}}$$

To verify that these roots are consistent with the assumption that $|A|, |C| \gg |B|$, we substitute these found roots back into $A$, $B$, and $C$ and compare their magnitudes. Using this method, we find that it is false that these roots are valid for small $\epsilon$, while validity for large $\epsilon$ is true. Therefore, these roots are valid in the limit of large positive $\epsilon$ only.

By the Fundamental Theorem of Algebra, a polynomial of degree 6.0 has exactly 6.0 roots. We have found 6.0 roots that are valid in the limit of small positive $\epsilon$ and 6.0 roots valid in the limit of large positive $\epsilon$. Our method therefore provides a complete solution to the problem, finding the correct number of roots in each $\epsilon$ regime.

The roots of $P(x)$ for large positive $\epsilon$ are

$$-\sqrt[6]{-\frac{1}{\epsilon}}, \quad \sqrt[6]{-\frac{1}{\epsilon}}, \quad \frac{\sqrt[6]{-\frac{1}{\epsilon}}\left(-1-\sqrt{3}i\right)}{2},$$

$$\frac{\sqrt[6]{-\frac{1}{\epsilon}}\left(-1+\sqrt{3}i\right)}{2}, \quad \frac{\sqrt[6]{-\frac{1}{\epsilon}}\left(1-\sqrt{3}i\right)}{2}, \quad \frac{\sqrt[6]{-\frac{1}{\epsilon}}\left(1+\sqrt{3}i\right)}{2}$$

and the roots of $P(x)$ for small positive $\epsilon$ are

$$\frac{1}{\epsilon}, \ 1, \ -\frac{1}{4}+\frac{\sqrt{5}}{4}-\frac{i\sqrt{2\sqrt{5}+10}}{4}, \ -\frac{1}{4}+\frac{\sqrt{5}}{4}+\frac{\sqrt{-10-2\sqrt{5}}}{4},$$

$$-\frac{\sqrt{5}}{4}-\frac{1}{4}-\frac{i\sqrt{10-2\sqrt{5}}}{4}, \ -\frac{\sqrt{5}}{4}-\frac{1}{4}+\frac{i\sqrt{10-2\sqrt{5}}}{4}$$

### A.2.3 POLYNOMIAL ROOT CORRECTION TERMS

The true roots $x^*$ of a polynomial are given by $x^*(\epsilon) = \overline{x}(\epsilon)+\delta$, where $\overline{x}$ is our existing approximation to the root as found in Appendix A.3 and $\delta$ is the error term. This requires us to solve

$$\epsilon(\overline{x}+\delta)^{n_1} \pm (\overline{x}+\delta)^{n_2} \pm 1 = 0$$

for $\delta$ by equating coefficients of $\epsilon$ terms of the same order, as detailed in the worked solution below.

---

**Sample Numeric Nondimensionalization Problem and Full Solution**

**Problem:** Consider the polynomial

$$P(x) = \epsilon x^3 - x + 1.$$

Find approximate expressions for all roots of the polynomial in the limit of small positive $\epsilon$ and large positive $\epsilon$. Use a series expansion to calculate improved formulae for these roots to order 1 i.e. calculate $\mathcal{O}(1)$ corrections for each root.

---

**Solution:** Note: The root calculation in this problem follow the same method as those demonstrated in the A.3, so they has been omitted here. We include only correction term calculations for the sake of brevity.

We now need to calculate correction terms for these roots to give us better approximations. We consider the ansatz that the root is given by $\overline{x} + \delta$, where the correction term $\delta$ is the sum of higher order terms of $\epsilon$ that we initially neglected in our approximation $\overline{x}$. By definition, $\delta < \overline{x}$. We plug this ansatz into the polynomial and perform a series expansion in $\delta$. We keep terms only up to $\mathcal{O}(1)$ in $\delta$. Then, we set the expression equal to 0 and solve for $\delta$.

Regime 1: valid for small $\epsilon$

Root 1: $-\sqrt{\frac{1}{\epsilon}}$

$$\overline{x} + \delta = -\sqrt{\frac{1}{\epsilon}} + \delta$$

Substitute this into $P(x)$ for $x$ and equate to 0:

$$-\delta + \epsilon\left(\delta - \sqrt{\frac{1}{\epsilon}}\right)^3 + \sqrt{\frac{1}{\epsilon}} + 1 = 0.$$

We then expand this expression to get

$$\delta^3\epsilon - 3\delta^2\epsilon\sqrt{\frac{1}{\epsilon}} + 2\delta - \epsilon\left(\frac{1}{\epsilon}\right)^{\frac{3}{2}} + \sqrt{\frac{1}{\epsilon}} + 1 = 0$$

and represent it as a series of $\mathcal{O}(1)$ in $\delta$, discarding higher order $\delta$ terms

$$2\delta - \epsilon \left(\frac{1}{\epsilon}\right)^{\frac{3}{2}} + \sqrt{\frac{1}{\epsilon}} + 1 \approx 0.$$

We can then solve the expression for the correction $\delta$ to $\mathcal{O}(1)$, and get

$$\boxed{\delta \approx \frac{\epsilon \left(\frac{1}{\epsilon}\right)^{\frac{3}{2}}}{2} - \frac{\sqrt{\frac{1}{\epsilon}}}{2} - \frac{1}{2}.}$$

Root 2: $\sqrt{\frac{1}{\epsilon}}$

$$\overline{x} + \delta = \sqrt{\frac{1}{\epsilon}} + \delta$$

Substitute this into $P(x)$ for $x$ and equate to 0:

$$-\delta + \epsilon \left(\delta + \sqrt{\frac{1}{\epsilon}}\right)^3 - \sqrt{\frac{1}{\epsilon}} + 1 = 0.$$

We then expand this expression to get

$$\delta^3 \epsilon + 3\delta^2 \epsilon \sqrt{\frac{1}{\epsilon}} + 2\delta + \epsilon \left(\frac{1}{\epsilon}\right)^{\frac{3}{2}} - \sqrt{\frac{1}{\epsilon}} + 1 = 0$$

and represent it as a series of $\mathcal{O}(1)$ in $\delta$, discarding higher order $\delta$ terms

$$2\delta + \epsilon \left(\frac{1}{\epsilon}\right)^{\frac{3}{2}} - \sqrt{\frac{1}{\epsilon}} + 1 \approx 0.$$

We can then solve the expression for the correction $\delta$ to $\mathcal{O}(1)$, and get

$$\boxed{\delta \approx -\frac{\epsilon \left(\frac{1}{\epsilon}\right)^{\frac{3}{2}}}{2} + \frac{\sqrt{\frac{1}{\epsilon}}}{2} - \frac{1}{2}.}$$

Regime 2: valid for small $\epsilon$
Root 1: 1

$$\overline{x} + \delta = 1 + \delta$$

Substitute this into $P(x)$ for $x$ and equate to 0:

$$-\delta + \epsilon \left(\delta + 1\right)^3 = 0.$$

We then expand this expression to get

$$\delta^3 \epsilon + 3\delta^2 \epsilon + 3\delta\epsilon - \delta + \epsilon = 0$$

and represent it as a series of $\mathcal{O}(1)$ in $\delta$, discarding higher order $\delta$ terms

$$\delta \left(3\epsilon - 1\right) + \epsilon \approx 0.$$

We can then solve the expression for the correction $\delta$ to $\mathcal{O}(1)$, and get

$$\boxed{\delta \approx -\frac{\epsilon}{3\epsilon - 1}.}$$

Regime 3: valid for large $\epsilon$

Root 1: $\sqrt[3]{-\frac{1}{\epsilon}}$

$$\overline{x} + \delta = \sqrt[3]{-\frac{1}{\epsilon}} + \delta$$

Substitute this into $P(x)$ for $x$ and equate to 0:

$$-\delta + \epsilon\left(\delta + \sqrt[3]{-\frac{1}{\epsilon}}\right)^3 - \sqrt[3]{-\frac{1}{\epsilon}} + 1 = 0.$$

We then expand this expression to get

$$\delta^3\epsilon + 3\delta^2\epsilon\sqrt[3]{-\frac{1}{\epsilon}} + 3\delta\epsilon\left(-\frac{1}{\epsilon}\right)^{\frac{2}{3}} - \delta - \sqrt[3]{-\frac{1}{\epsilon}} = 0$$

and represent it as a series of $\mathcal{O}(1)$ in $\delta$, discarding higher order $\delta$ terms

$$\delta\left(3\epsilon\left(-\frac{1}{\epsilon}\right)^{\frac{2}{3}} - 1\right) - \sqrt[3]{-\frac{1}{\epsilon}} \approx 0.$$

We can then solve the expression for the correction $\delta$ to $\mathcal{O}(1)$, and get

$$\boxed{\delta \approx \frac{\sqrt[3]{-\frac{1}{\epsilon}}}{3\epsilon\left(-\frac{1}{\epsilon}\right)^{\frac{2}{3}} - 1}.}$$

Root 2: $\frac{\sqrt[3]{-\frac{1}{\epsilon}}\left(-1 - \sqrt{3}i\right)}{2}$

$$\overline{x} + \delta = \frac{\sqrt[3]{-\frac{1}{\epsilon}}\left(-1 - \sqrt{3}i\right)}{2} + \delta$$

Substitute this into $P(x)$ for $x$ and equate to 0:

$$-\delta + \epsilon\left(\delta + \frac{\sqrt[3]{-\frac{1}{\epsilon}}\left(-1 - \sqrt{3}i\right)}{2}\right)^3 - \frac{\sqrt[3]{-\frac{1}{\epsilon}}\left(-1 - \sqrt{3}i\right)}{2} + 1 = 0.$$

We then expand this expression to get

$$\delta^3\epsilon - \frac{3\delta^2\epsilon\sqrt[3]{-\frac{1}{\epsilon}}}{2} - \frac{3\sqrt{3}i\delta^2\epsilon\sqrt[3]{-\frac{1}{\epsilon}}}{2} - \frac{3\delta\epsilon\left(-\frac{1}{\epsilon}\right)^{\frac{2}{3}}}{2}$$
$$+ \frac{3\sqrt{3}i\delta\epsilon\left(-\frac{1}{\epsilon}\right)^{\frac{2}{3}}}{2} - \delta + \frac{\sqrt[3]{-\frac{1}{\epsilon}}}{2} + \frac{\sqrt{3}i\sqrt[3]{-\frac{1}{\epsilon}}}{2} = 0$$

and represent it as a series of $\mathcal{O}(1)$ in $\delta$, discarding higher order $\delta$ terms

$$\delta\left(-\frac{3\epsilon\left(-\frac{1}{\epsilon}\right)^{\frac{2}{3}}}{2} + \frac{3\sqrt{3}i\epsilon\left(-\frac{1}{\epsilon}\right)^{\frac{2}{3}}}{2} - 1\right) + \frac{\sqrt[3]{-\frac{1}{\epsilon}}}{2} + \frac{\sqrt{3}i\sqrt[3]{-\frac{1}{\epsilon}}}{2} \approx 0.$$

We can then solve the expression for the correction $\delta$ to $\mathcal{O}(1)$, and get

$$\boxed{\delta \approx \frac{\sqrt[3]{-\frac{1}{\epsilon}}\left(1 + \sqrt{3}i\right)}{3\epsilon\left(-\frac{1}{\epsilon}\right)^{\frac{2}{3}} - 3\sqrt{3}i\epsilon\left(-\frac{1}{\epsilon}\right)^{\frac{2}{3}} + 2}.}$$

Root 3: $\frac{\sqrt[3]{-\frac{1}{\epsilon}}\left(-1 + \sqrt{3}i\right)}{2}$

$$\overline{x} + \delta = \frac{\sqrt[3]{-\frac{1}{\epsilon}}\left(-1 + \sqrt{3}i\right)}{2} + \delta$$

Substitute this into $P(x)$ for $x$ and equate to $0$:

$$-\delta + \epsilon \left(\delta + \frac{\sqrt[3]{-\frac{1}{\epsilon}}\left(-1+\sqrt{3}i\right)}{2}\right)^3 - \frac{\sqrt[3]{-\frac{1}{\epsilon}}\left(-1+\sqrt{3}i\right)}{2} + 1 = 0.$$

We then expand this expression to get

$$\delta^3 \epsilon - \frac{3\delta^2\epsilon\sqrt[3]{-\frac{1}{\epsilon}}}{2} + \frac{3\sqrt{3}i\delta^2\epsilon\sqrt[3]{-\frac{1}{\epsilon}}}{2} - \frac{3\delta\epsilon\left(-\frac{1}{\epsilon}\right)^{\frac{2}{3}}}{2}$$
$$-\frac{3\sqrt{3}i\delta\epsilon\left(-\frac{1}{\epsilon}\right)^{\frac{2}{3}}}{2} - \delta + \frac{\sqrt[3]{-\frac{1}{\epsilon}}}{2} - \frac{\sqrt{3}i\sqrt[3]{-\frac{1}{\epsilon}}}{2} = 0$$

and represent it as a series of $\mathcal{O}(1)$ in $\delta$, discarding higher order $\delta$ terms

$$\delta\left(-\frac{3\epsilon\left(-\frac{1}{\epsilon}\right)^{\frac{2}{3}}}{2} - \frac{3\sqrt{3}i\epsilon\left(-\frac{1}{\epsilon}\right)^{\frac{2}{3}}}{2} - 1\right) + \frac{\sqrt[3]{-\frac{1}{\epsilon}}}{2} - \frac{\sqrt{3}i\sqrt[3]{-\frac{1}{\epsilon}}}{2} \approx 0.$$

We can then solve the expression for the correction $\delta$ to $\mathcal{O}(1)$, and get

$$\boxed{\delta \approx \frac{\sqrt[3]{-\frac{1}{\epsilon}}\left(1-\sqrt{3}i\right)}{3\epsilon\left(-\frac{1}{\epsilon}\right)^{\frac{2}{3}}+3\sqrt{3}i\epsilon\left(-\frac{1}{\epsilon}\right)^{\frac{2}{3}}+2}.}$$

### A.2.4  ODEs

We generate third-order ordinary differential equations of the form

$$y''' = f_1(x)(y'')^a + f_2(x)(y')^b + f_3(x)y^c + f_4(x),$$

where $f_1(x), f_2(x), f_3(x), f_4(x)$ are rational functions with integer coefficients. The initial conditions are randomly selected integers from $[0, 3]$. The dataset excludes problems with a function of $x$ as a dominant term because of the difficulty of deriving power law expressions in these cases.

Approximate solutions at small $x$ can be derived using a Taylor series expansion (up to the third order) around $x = 0$. Solving ODEs in the large $x$ regime involves determining the two largest terms, assuming a divergence at some large $x^*$, and solving the dominant balance between these terms to create a power law approximation of the form

$$y(x) = A(x^* - x)^p.$$

---

**ODE Problem and Solution**

Problem: Consider the following third-order ordinary differential equation:

$$y''' = -\frac{y}{24x^4 + 6x^2 + 3} + y'^2 - \frac{y''}{5x^3 - 2x^2 - x + 2} - \frac{1}{12x^2 - \cos(x) + 11}$$

with initial conditions at  x = 0:

$$y(0) = 1.00$$
$$y'(0) = 0.00$$
$$y''(0) = 0.00$$

Find analytical expressions that approximate the solution of y(x) at small and large $x$.

**Solution:**

---

The dominant balance in the large x regime is given by

$$\frac{d^3}{dx^3}y = \left(\frac{d}{dx}y\right)^2.$$

We recognize that the solution of this ODE will diverge at finite $x$ and that divergences typically follow a power law of the form

$$y = \alpha(x - x^*)^p,$$

where $x^*$ is the divergence point. The divergence point can be determined by estimated by examining the numerical solution generated by code.

Plugging in the dominant terms we found previously yields the following equation:

$$\alpha p\,(p-2)\,(p-1)\,(x-11.45)^{p-3} = \alpha^2 p^2\,(x-11.45)^{2p-2}.$$

After substituting the derivatives, the equation is reorganized to collect terms with respect to $(x - x^*)$. This leads to an equation where the coefficients and powers of $(x - x^*)$ are equated on both sides. Simplifying the equation gives us two separate equations, one for the coefficients and another for the powers of $(x - x^*)$. There is now a system of equations, where the coefficients' equation is

$$\alpha p\,(p-2)\,(p-1) = \alpha^2 p^2$$

and the powers' equation is:

$$p - 3 = 2p - 2.$$

Solving this system of equations provides the values of $\alpha$ and $p$. A valid solution is identified if $\alpha$ and $p$ are both nonzero. Here, the solution for $\alpha$ and $p$ is found to be:

$$\alpha = -6, \quad p = -1$$

With these values, the analytical approximation for the solution at large $x$ (near the divergence point) is given by

$$y = -6(x - 11.45)^{-1}.$$

The approximate solution at small $x$ can also be solved used dominant balance, but one can take advantage of the initial conditions and form a Taylor series instead around $x = 0$, which is given by

$$y(x) \approx y(0) + y'(0)x + \frac{y''(0)}{2!}x^2 + \frac{y'''(0)}{3!}x^3.$$

Plugging in the initial conditions, we get the following expression at small $x$:

$$y(x) = 1 - \frac{13}{180}x^3$$

Thus, with rounding for clarity, the solution is given by

$$\boxed{y(x) = 1 - \frac{13}{180}x^3, \ y = -6(x - 11.45)^{-1}.}$$

### A.2.5 INTEGRALS

The polynomial $P(x)$ is randomly generated to consist of up to ten terms, where each term is a power function of $x$ with an integer power randomly sampled from 1 and 20 and an integer coefficient sampled from 1 to 10. The integration bound $a \in [0, 100]$ is also randomly selected. This form ensures that the integral does not oscillate.

The height is approximated as the maximum value of the integrand, which is $\frac{1}{\epsilon}$, and the width can be estimated as the distance over which the integrand decreases from its maximum value by a factor of 2, which implies that the width $x$ obeys the equation

$$\frac{1}{\epsilon + P(x)} = \frac{1}{2\epsilon} \Rightarrow P(x) = \epsilon.$$

In the regime of small $\epsilon$, the term with the smallest degree and $\epsilon$ are the dominant terms, and in the regime of intermediate $\epsilon$, the term with the largest degree and $\epsilon$ are dominant. There exists one more solution regime when the width of the integral exceeds the limits of integration, or when $\epsilon$ is "very large." In this case, the integral is approximated by $L/\epsilon$, where $L$ is the integration range.

See a full sample of a traditional integral problem in Box 1.

### A.2.6 LAPLACE INTEGRALS

Laplace integrals of the form $I(x) = \int_a^b g(t)e^{\pm x f(t)} dt$ assume that $f(t) > 0$, is never a constant, and has an absolute minimum at a point $t_0$ either in the interior of or on the bounds of the interval $[a, b]$.

The set of possible Laplace integrals $I(x)$ in our dataset are parameterized by four parameters: the bounds $[a, b]$, $g(t)$, $f(t)$, and the sign in front of $x$. To generate the dataset, the bounds for each problem were randomly sampled from the $[-1, -0.9, \ldots 0.9, 1]$, and the sign was uniformly sampled from $\{-1, 1\}$. The functions $f(t)$ and $g(t)$ were generated by randomly selecting a linear combination of polynomials up to fifth order and basic trigonometric functions.

Our solution uses SymPy under the hood to find the minima of $f(t)$ (or the dual annealing algorithm if SymPy fails to return the minima).

---

**Sample Laplace Integral Problem and Solution**

Problem: Consider the integral

$$I(x) = \int_{-0.9}^{0.3} (-1.6t^2 - 0.5\sin(t) - 1.9)e^{+x(-2.5t^4 - 0.8t^3 + 1.4t^2)} dt \qquad (2)$$

Develop an analytical formula for $I(x)$ that is accurate as $x \to \infty$.

---

**Solution:**
The integral is of the form

$$I(x) = \int_a^b g(t)e^{+x f(t)} dt \qquad (3)$$

where $a = -0.9$, $b = 0.3$, $g(t) = -1.6t^2 - 0.5\sin(t) - 1.9$, and $f(t) = -2.5t^4 - 0.8t^3 + 1.4t^2$. This means we can use Laplace's method to develop an analytical approximation in the limit that $x \to \infty$. In this limit, the integral will be dominated by the integrand near the maximum of $f(t)$ within the bounds $[-0.9, 0.3]$. So, to simplify the integral, we will expand the integrand around this maximum. In this case, we can find the maximum of $f(t) = -2.5t^4 - 0.8t^3 + 1.4t^2$ on the interval analytically. We begin by looking for critical point(s) $t_{crit}$ of $f(t)$ by solving $f'(t) = -10.0t^3 - 2.4t^2 + 2.8t = 0$ for $t$. This gives us that $t_{crit} = [-0.66, 0]$. To find the maximum on this interval, we evaluate $f(t)$ at the critical point(s) $t_{crit}$ and the bounds $-0.9$ and $0.3$. We take the $t$ that gives the largest value. Here, this maximum $t_0 = [-0.66]$. Since the integral is dominated by the value of the integrand near -0.66, we Taylor expand the integrand around this point.

$$I(x) = \int_a^b (g(-0.66) + (t + 0.66)g'(-0.66) + \ldots)$$
$$* e^{+x(f(-0.66) + (t+0.66)f'(-0.66) + \frac{(t+0.66)^2}{2}f''(-0.66) + \ldots)} dt \qquad (4)$$

But $f'(-0.66) = 0$ by definition, so we can remove this term from the exponent. We can then approximate

$$I(x) \approx \int_a^b g(-0.66)e^{+x(f(-0.66) + \frac{(t+0.66)^2}{2}f''(-0.66))} dt, \qquad (5)$$

which equals

$$g(-0.66)e^{+x f(-0.66)} \int_a^b e^{+x(\frac{(t+0.66)^2}{2}f''(-0.66))} dt \qquad (6)$$

---

We perform the change of variables $u = \sqrt{x \frac{|f''(-0.66)|}{2}}(t + 0.66)$, rewriting the integral as

$$g(-0.66)e^{+xf(-0.66)} \int_{\sqrt{x \frac{|f''(-0.66)|}{2}}(a+0.66)}^{\sqrt{x \frac{|f''(-0.66)|}{2}}(b+0.66)} \sqrt{\frac{2}{x|f''(-0.66)|}} e^{-u^2} \, dt \tag{7}$$

Since $x \to \infty$, we approximate this as

$$g(-0.66)e^{+xf(-0.66)} \sqrt{\frac{2}{x|f''(-0.66)|}} \int_{-\infty}^{\infty} e^{-u^2} \, dt \tag{8}$$

Solving the integral and evaluating, we find that

$$\boxed{I(x) \approx -1.21 \sqrt{\frac{\pi}{x}} e^{0.37x}} \tag{9}$$

### A.2.7 WORD PROBLEM AUTOMATIC GENERATION SETUP

Table 3: Prompts for generating problem and solution contexts and plausibility verification.

| Task | Task instruction |
|------|------------------|
| Question contextualization | Rewrite this `{question_type}` problem by embedding it within a plausible real-world `{domain_seed}` problem scenario, without changing the mathematical question at the end. |
| Solution contextualization | We provide above a real-world `{domain_seed}` question, and its mathematical solution. Generate a single introductory sentence before the solution that can connect the solution to the real-world `{domain_seed}` context provided in the question. |
| Verification | We provide above a real-world `{domain_seed}` question. Review and verify its plausibility within the context of parameter ranges of this domain (e.g. energy value cannot be negative). Identify any inconsistencies or areas needing clarification to ensure the problem is realistic for this domain. Return a single float number as plausibility score (0-1) in LaTeX boxed format `\boxed{}` at the end. |

### A.2.8 WORD PROBLEM AUTOMATIC GENERATION EXAMPLE

---

**Original vs. physics context augmented problem (*ODEs*)**

**Original Mathematical Problem:** Consider the following third-order ordinary differential equation:

$$y''' = \frac{y}{3x^4 - 4x^3 - 2} + (y')^4 + \frac{y''}{4 \cdot (4x^2 + 1)} + \frac{1}{5x^4 + \sin(x) - 8}$$

with initial conditions at $x = 0$:

$$y(0) = 1.00$$
$$y'(0) = 0.00$$
$$y''(0) = 0.00$$

Find analytical expressions that approximate the solution of $y(x)$ in the small $x$ and large $x$ regimes.

---

**Real-World Context Problem:** In a study of the dynamics of a newly discovered type of fluid, researchers are investigating the behavior of a fluid flow through a porous medium. The flow is described by a third-order ordinary differential equation that models the velocity profile of the fluid, denoted by $y(x)$, as it moves through the medium. The equation takes into account various factors such as the porosity of the medium, the viscosity of the fluid, and external forces acting on the fluid.
The governing equation for the velocity profile is given by:

$$y''' = \frac{y}{3x^4 - 4x^3 - 2} + (y')^4 + \frac{y''}{4 \cdot (4x^2 + 1)} + \frac{1}{5x^4 + \sin(x) - 8}$$

The initial conditions at the entry point of the medium, $x = 0$, are specified as follows:

$$y(0) = 1.00$$
$$y'(0) = 0.00$$
$$y''(0) = 0.00$$

---

> The researchers are interested in finding analytical expressions that approximate the solution of $y(x)$ in the small $x$ and large $x$ regimes. These approximations will help in understanding the initial behavior of the fluid as it enters the medium and its asymptotic behavior as it travels further through the medium.

### A.2.9 WORD PROBLEM AUTOMATIC GENERATION QUALITY EVALUATION

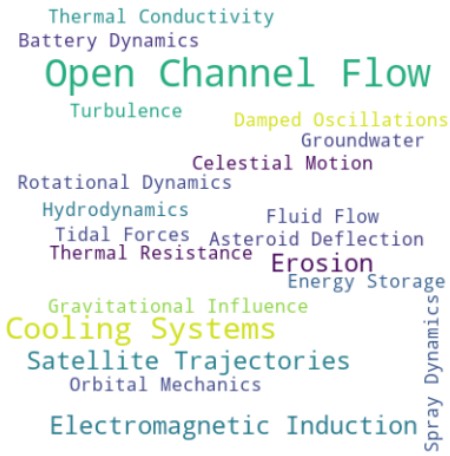
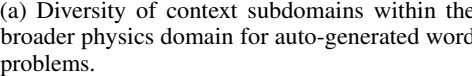

(a) Diversity of context subdomains within the broader physics domain for auto-generated word problems.

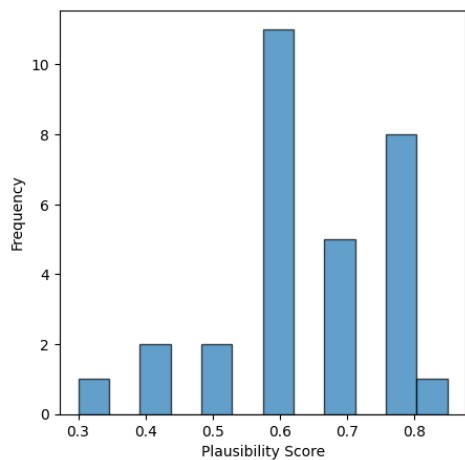

(b) Plausibility scores distribution of automatically generated word problems.

Figure 6: Diversity and quality assessment of automatically generated word problems. Only problems above a plausibility score threshold are included.

A.3 EVALUATION SETUP

A.3.1 PROMPTS FOR RESPONSE GENERATION

Table 4: Problem type specific hints by Question and Answer Type

| Question Type | Answer Type | Task instruction |
| --- | --- | --- |
| Nondim-symbolic | SymPy | Please answer the question requiring an answer in a SymPy convertible formula containing variables and math operation expressions and provide the final answer, e.g., $x^3$, $\frac{x}{y}$ inside a Latex boxed format \boxed{}. |
| Nondim-numerical | Float (2) | Please answer the question requiring a floating-point number with two decimal places and provide the final value, e.g., 0.80, 3.12, inside a Latex box \boxed{}. |
| Polynomial Roots | SymPy List | Please answer the question requiring a Python list containing SymPy convertible formulas of variable $\epsilon$ and math operation expressions and provide the final list, e.g., $[\epsilon^3, \frac{1}{\epsilon}]$ inside a Latex boxed format \boxed{}. |
| ODEs | SymPy List | Please answer the question requiring a Python list containing SymPy convertible formula of $y = f(x)$ and provide the final list, e.g., $[y = 1 - x^3, y = -6/(x - 5)]$, inside a Latex boxed format \boxed{}. |
| Integrals | SymPy | Please answer the question requiring an answer in a SymPy convertible formula containing formulas of variable $x$ and math operation expressions and provide the final answer, e.g., $x^3$ inside a Latex boxed format \boxed{}. |

### A.3.2 Prompts for grading

Table 5: LLM-based grading prompts by Question and Answer Type

| Question type | Answer type | Task instruction |
|---|---|---|
| Polynomial Roots | SymPy List | Please take this response `{response}` and this answer key `{answer key}` and grade the response based on the following criteria: 1) Check both the small and large $\epsilon$ solutions. 2) For each solution, give full credit if it completely matches the elements in the answer key; give partial credit proportional to the number of matching roots between the response and the answer key; give no credit if it is completely wrong. 3) For both partial and no credit briefly state the error reason. 4) Average the scores for the small and large epsilon solutions to obtain a final score between 0 and 1. 5) Give the final grading as a float in Latex boxed format `\boxed{}`. |
| ODEs | SymPy List | Please take this response `{response}` and this solution `{answer key}` and grade the response based on the following criteria: 1) Check both the small and large $\epsilon$ solutions. 2) For small regime solution, only give full credit if it matches the formula in the answer key exactly; give no credit if it is doesn't match the form. For large regime solution, give full credit if it matches the formula in the answer key exactly; give partial credit if it doesn't match but the numerical evaluation is not far from solution at this regime; give no credit if neither satisfies 3) Average the scores for the small and large epsilon solutions to obtain a final score between 0 and 1. 4) Give the final grading as a float in Latex boxed format `\boxed{}`. |
| Integrals (traditional) | SymPy List | Please take this response `{response}` and this solution `{answer key}` and grade the response based on the following criteria: 1) Check both the small and large $\epsilon$ solutions. 2) For each solution, give full credit if it matches the formula in the answer key; give no credit if it is completely wrong and briefly state the reason for the error. 3) Average the scores for the small and large epsilon solutions to obtain a final score between 0 and 1. 4) Give the final grading as a float in Latex boxed format `\boxed{}`. |
| Integrals (Laplace) | SymPy | Please take this response `{response}` and this solution `{answer key}` and grade the response based on the following criteria: 1) Check the large $x$ final solution. 2) Give full credit if it matches the formula in the answer key; give half credit if the `{response}` get to the checkpoint where it correctly identifies $t_0$ where $f$ attains its maximum and attempt performing Taylor's expansion around it but the final answer is wrong; give no credit if it is completely wrong. 3) For both partial and no credit briefly state the error reason. 4) Give the final grading as a float in Latex boxed format `\boxed{}`. |

### A.3.3 GPT GRADING HUMAN VERIFICATION

| Model | Roots | ODEs | Integrals |
|---|---|---|---|
| GPT3.5 (0) | 0 | 0 | 0 |
| GPT3.5 (1) | 0 | -0.09 | -0.02 |
| GPT3.5 (5) | +0.02 | +0.07 | +0.02 |
| GPT4 (0) | 0 | -0.02 | 0 |
| GPT4 (1) | 0 | -0.04 | -0.02 |
| GPT4 (5) | +0.07 | -0.07 | -0.15 |
| o1-mini (0) | +0.04 | +0.05 | 0 |
| o1-mini (5) | +0.05 | +0.05 | 0 |
| Llama3-8b (0) | 0 | 0 | -0.02 |
| Llama3-8b (5) | -0.07 | -0.02 | -0.02 |
| Codellama3-14b (0) | 0 | -0.02 | 0 |
| Codellama3-14b (5) | 0 | -0.02 | 0 |

Table 6: Average adjusted points using human judgment from GPT-based grading. Rows with score adjustments of 0.1 or more are highlighted in pink.

### A.3.4 MODEL HYPER-PARAMETERS

Table 7: Generating parameters for various LLMs.

| Model | Generation Setup |
|---|---|
| GPT-3.5 | `model = gpt-3.5-turbo, temperature = 0, max_tokens = 4000` |
| GPT-4 | `model = gpt-4-turbo, temperature = 0, max_tokens = 4000` |
| o1-mini | `model = o1-mini, temperature = 0, max_tokens = 4000` |
| Llama3 | `model = llama3:8b, temperature = 0` |
| CodeLlama | `model = codellama:13b, temperature = 0` |

### A.3.5 COMPUTING RESOURCE

Evaluations of open-source models on **HARDMATH** are conducted on a high-performance compute cluster with a single Tesla V100 GPU (16GB vram). Evaluation on one problem type typically takes less than 1 hour.

## A.4 EXTENDED EXPERIMENTAL RESULTS

### A.4.1 EXTENDED EVALUATION RESULTS

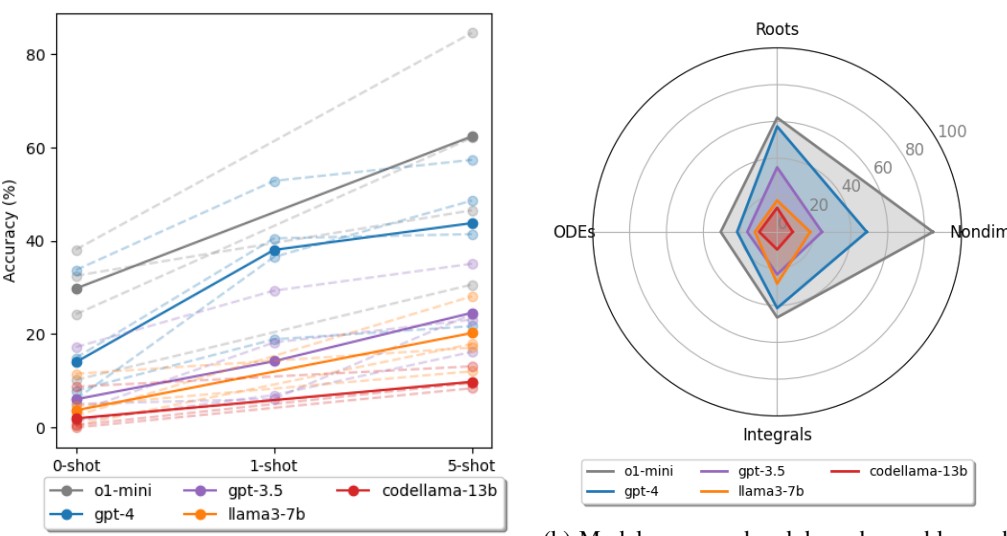

(a) Model accuracy increases with shot numbers.

(b) Model accuracy breakdown by problem sub-types for all models with 5-shot CoT prompting.

Figure 7: Model evaluation accuracy breakdown by shot number and problem sub-types. (a) evaluation accuracy for all models increases with shot numbers for CoT prompting with o1-mini and GPT-4 showing the most obvious improvements; (b) evaluation accuracy breakdown for all models on all problem sub-types under the 5-shot CoT condition.

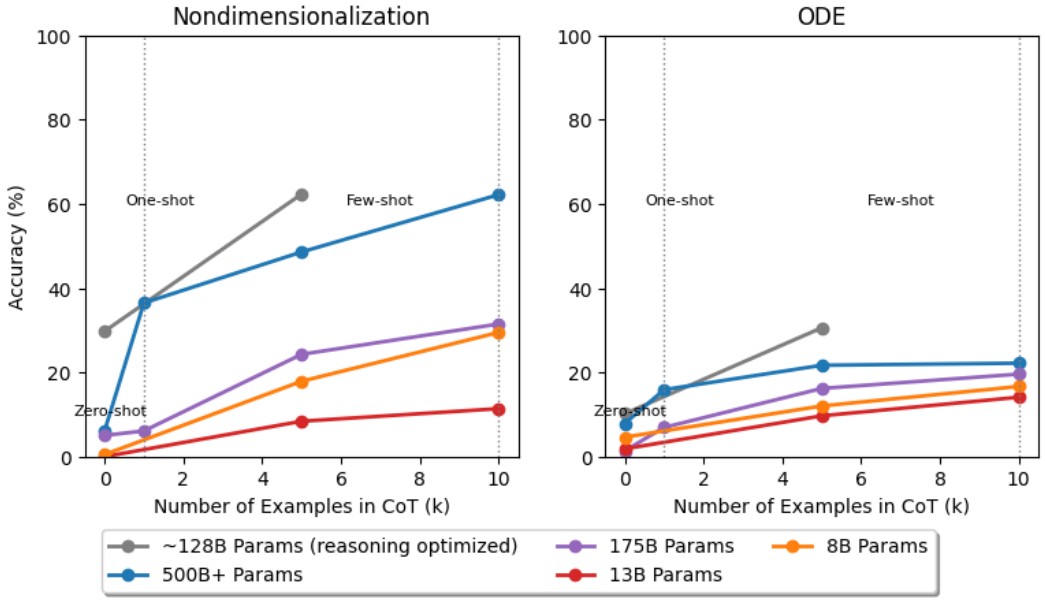

Figure 8: Model performance scaling upon few-shot prompting (0, 1, 5, 10) for problem type *Nondim* and *ODEs*

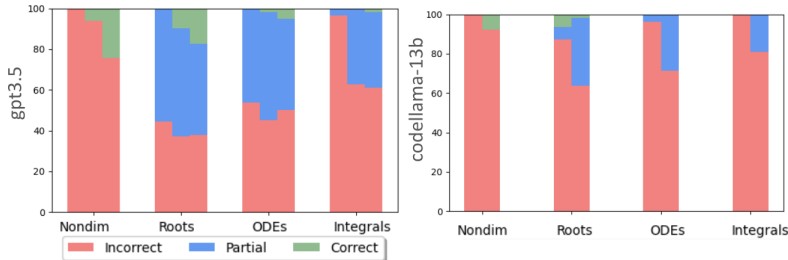

Figure 9: Percentage of correct, partial, and incorrect responses for GPT-3.5 and CodeLlama using CoT.

### A.4.2 EXTENDED ERROR MODE ANALYSIS

---

**Problem 136 Grading Response:** *Roots* (error mode: incorrect dominant balance setup)

**Question:** Consider the polynomial $P(x) = \epsilon x^6 + x^4 - 1$. Find approximate expressions for all roots of the polynomials in the limit of small positive $\epsilon$ and large positive $\epsilon$. Only a single term approximation to the root is required.

**Grading for Large Positive $\epsilon$:**

**Model Response:** For large $\epsilon$ When $\epsilon$ is large, the term $\epsilon x^6$ dominates the polynomial for most $x$, except for potentially small $x$ where $x^4$ might still be relevant. However, focusing on the dominant term gives: $P(x) \approx \epsilon x^6$. Setting this equal to zero for an approximation (acknowledging that we're looking for non-trivial roots of the overall polynomial, but focusing on dominance to guide our approximation), suggests that we don't get meaningful roots from this term alone since $x^6 = 0$ doesn't provide roots that satisfy the original polynomial's requirement for non-zero roots.

**Reference Solution:** Finally, we examine the balance $A + C = 0$, assuming that $|B|$ is negligible when compared to $|A|$ and $|C|$. Solving this for $x$ in terms of $\epsilon$ gives us 6 non-zero roots: $\epsilon x^6 - 1 = 0$ implies $x = \left[ \dfrac{(-1 \pm \sqrt{3}i) \sqrt[6]{\frac{1}{\epsilon}}}{2}, \dfrac{(1 \pm \sqrt{3}i) \sqrt[6]{\frac{1}{\epsilon}}}{2}, -\sqrt[6]{\frac{1}{\epsilon}}, \sqrt[6]{\frac{1}{\epsilon}} \right]$. To verify the roots found for consistency with our initial assumption, we check that $|A|, |C| >> |B|$ holds for this root by substituting our found root expression for $x$ back into $A$, $B$, and $C$ and comparing their magnitudes. Using this method, we find that validity for small $\epsilon$ is False and validity for large $\epsilon$ is True.

**Grading:** The response wrongly sets up the dominant balance as $A = 0$ and does not provide a concrete list of roots for large $\epsilon$. Therefore, score for large positive $\epsilon$ is $\boxed{0.0}$

---

**Problem 133 Grading Response:** *Roots* (error mode: drop imaginary roots)

**Question:** Consider the polynomial $P(x) = \epsilon x^3 + x^2 - 1$. Find approximate expressions for all roots of the polynomials in the limit of small positive $\epsilon$ and large positive $\epsilon$. Only a single term approximation to the root is required.

**Grading for Large Positive $\epsilon$:**

**Model Response:** To find the approximate expressions for all roots of the polynomial $P(x) = \epsilon x^3 + x^2 - 1$ in the limits of small and large $\epsilon$, we will consider the two cases separately. For Large $\epsilon$ When $\epsilon$ is large, the term $\epsilon x^3$ dominates the polynomial. Thus, the polynomial can be approximated by ignoring the $x^2$ term, leading to $\epsilon x^3 - 1 \approx 0$. Solving for $x$ gives us an approximation for the root: $x \approx \left(\frac{1}{\epsilon}\right)^{\frac{1}{3}}$. The other roots in this limit are complex and given the scope, we'll focus on this real root approximation.

**Reference Solution:** Finally, we examine the balance $A + C = 0$, assuming that $|B|$ is negligible when compared to $|A|$ and $|C|$. Solving this for $x$ in terms of $\epsilon$ gives us 3 non-zero roots:

---

$\epsilon x^3 - 1 = 0$ implies $\boxed{x = \left[ \dfrac{(-1 - \sqrt{3}i) \sqrt[3]{\frac{1}{\epsilon}}}{2}, \dfrac{(-1 + \sqrt{3}i) \sqrt[3]{\frac{1}{\epsilon}}}{2}, \sqrt[3]{\dfrac{1}{\epsilon}} \right]}$. To verify the

roots found for consistency with our initial assumption, we check that $|A|, |C| >> |B|$ holds for this root by substituting our found root expression for $x$ back into $A$, $B$, and $C$ and comparing their magnitudes. Using this method, we find that validity for small $\epsilon$ is False and validity for large $\epsilon$ is True. Therefore, these roots are valid in the limit of large positive $\epsilon$ only.
**Grading**: The response makes additional assumptions to drop the two other complex roots for large $\epsilon$ regime without the question asking it to do so. Therefore, score for large positive $\epsilon$ is $\boxed{0.5}$

### A.4.3 EXTENDED EXAMPLE GPT-4O GRADING RESPONSES

---

**Grading Response: *ODEs* (incorrect)**

To grade the response, we will compare the provided solutions in the response with the ground truth solutions in both the small $x$ and large $x$ regimes.

#### SMALL $x$ REGIME

Response: $y(x) = 1 - 5e^{-x}$
Ground Truth: $y(x) = 1 - \frac{5x^3}{6}$
The response and the ground truth solution for small $x$ are significantly different. The ground truth solution is a polynomial in $x$, while the response includes an exponential term. They do not match at all, so the solution for the small $x$ regime is completely wrong.

#### LARGE $x$ REGIME

Response: $y(x) =$ "Assumed rapid growth form due to $y'^4$ term."
Ground Truth: $y = \sqrt[3]{30}(x - 2.68)^{\frac{1}{3}} + (1.84)$
The response does not provide a specific analytical expression for the large $x$ regime, while the ground truth does provide a specific form. The response mentions rapid growth due to $y'^4$, but without a clear analytical form, this response doesn't match the ground truth solution. This indicates that the solution for the large $x$ regime is also completely wrong.
Given the evaluation criteria, both solutions do not match or are incomplete compared to the ground truth. Therefore, each solution gets no credit.
The final grading is $\boxed{0.0}$.

---

Box 3: Example grading response for a *ODEs* problem that gets zero point.

---

**Grading Response: *Integrals* (correct)**

The model response approximates the integral for small $\epsilon$ as $\boxed{\dfrac{1}{6^{1/4}}\epsilon^{-3/4}}$ and for large $\epsilon$ as

$\boxed{\dfrac{1}{8^{1/18}}\epsilon^{-17/18}}$.

Now, let's compare these results with the ground truth solutions:

---

**For small $\epsilon$:** - The ground truth solution is $\boxed{I(\epsilon) = \dfrac{0.64}{\epsilon^{0.75}}}$. - The model's result is

$\boxed{\dfrac{1}{6^{1/4}}\epsilon^{-3/4}}$.

Both results for small $\epsilon$ are essentially the same, with:

$$\frac{1}{6^{1/4}} \approx 0.64$$

**For large $\epsilon$:** - The ground truth solution is $\boxed{I(\epsilon) = \dfrac{0.89}{\epsilon^{0.94}}}$. - The model's result is

$\boxed{\dfrac{1}{8^{1/18}}\epsilon^{-17/18}}$.

Both results for large $\epsilon$ are essentially the same, with:

$$\frac{1}{8^{1/18}} \approx 0.89$$

Since the model's response closely matches the ground truth solutions in both regimes (small and large $\epsilon$) and provides essentially the same approximations, I would give full credit $\boxed{1.0}$

Box 4: Example grading response for a *Integrals* sub-type simple integral problem that gets full points.

---

**Grading Response:** *Integrals* (partial)

**Answer Key**
1. Setup and Formulation: it 1) correctly identifies the integral form; 2) uses Laplace's method; 3) identifies critical points and uses Taylor expansion around the point of interest; 4) performs a change of variables to simplify the integral.
2. Key Steps and Approximations: it 1) expands the integrand around the minimum of $f(t)$; 2) approximates the integral by focusing on the dominant exponential term; 3) provides a final analytical approximation for $I(x)$ as $x \to \infty$.
3. Final Result:

$$\boxed{I(x) \approx -\frac{0.004e^{0.89x}}{x}}$$

**Model Answer**
1. Setup and Formulation: it 1) correctly identifies the integral form; 2) recognizes that the exponential term will dominate for large $x$; 3) focuses on the region where the integrand is most significant (near $t = 0.7$).
2. Key Steps and Approximations: it 1) uses the method of steepest descents heuristically without detailed Taylor expansion; 2) approximates by evaluating the polynomial part at $t = 0.7$ and considering the exponential decay; 3) recognizes the dominant behavior of the exponential term but does not perform a detailed variable change or expansion.
3. Final Result:

$$\boxed{I(x) \approx -0.017 \cdot e^{x \cdot 2.6 \cdot 0.7^3}}$$

**Comparison**
1. Similarities: 1) understanding of dominance: both the answer key and the student correctly identify that the behavior of the integral for large $x$ is dominated by the exponential term; 2) identifying significant points: both recognize that $t = 0.7$ is the critical point where the integrand contributes significantly; 3) approximate behavior: Bbth solutions focus on

approximating the behavior of the integral for large $x$, emphasizing the exponential term's dominance.

2. Differences: 1) mathematical rigor: the answer key employs a more rigorous mathematical approach, including Taylor expansion and change of variables, providing a more precise and detailed solution; 2) heuristic approach: the student takes a more heuristic approach, focusing on the dominant exponential term without detailed mathematical expansions or variable changes; 3) accuracy of final expression: the answer key provides a more accurate final expression with specific constants derived from the detailed process, while the student's result, though on the right track, has a different coefficient due to the heuristic method. I would give partial credit $\boxed{0.5}$

Box 5: Example grading response for a *Integrals* sub-type Laplace integral problem highlighting the error of failing to develop Taylor's expansion.

## A.5 MOTIVATION FOR INCLUDING NONDIMENSIONALIZATION PROBLEMS

> **GPT-4 response on solving a dimensionalized polynomial root finding problem**
>
> **Problem:** Find approximate analytical expressions for the roots of the polynomial:
> $$a_0 x^5 - a_1 x + a_2 = 0$$
> at different regimes of the coefficients $a_0$, $a_1$, and $a_2$.
>
> ---
>
> **Model Solution:**
> Regime 1: $|a_0| \gg |a_1|, |a_2|$ (Dominance of the $x^5$-term) In this regime, the polynomial simplifies to:
> $$a_0 x^5 \approx 0$$
> Hence, the dominant contribution to the roots arises from balancing $a_0 x^5$ with the constant term $a_2$:
> $$a_0 x^5 + a_2 \approx 0 \implies x \approx \left(-\frac{a_2}{a_0}\right)^{1/5}.$$
> This gives five distinct roots corresponding to the 5$^{\text{th}}$ roots of unity:
> $$x_k = \left(-\frac{a_2}{a_0}\right)^{1/5} e^{i2\pi k/5}, \quad k = 0, 1, 2, 3, 4.$$
>
> Regime 2: $|a_1| \gg |a_0|, |a_2|$ (Dominance of the linear term) Here, the polynomial reduces to:
> $$-a_1 x \approx 0$$
> and the root is approximately:
> $$x \approx 0.$$
> Including the next dominant term, we balance $-a_1 x$ with $a_2$:
> $$-a_1 x + a_2 \approx 0 \implies x \approx \frac{a_2}{a_1}.$$
>
> Regime 3: $|a_2| \gg |a_0|, |a_1|$ (Dominance of the constant term) In this regime, the constant term dominates, and the polynomial becomes inconsistent unless $a_2 = 0$, which contradicts the assumption $|a_2| \neq 0$. Thus, there are no real roots in this regime.
>
> Regime 4: General Balance of Terms When all terms are of comparable magnitude, we consider approximate solutions based on the magnitude of $x$:
>
> 1. **For large $x$:** The $x^5$-term dominates. Balancing $a_0 x^5$ with $-a_1 x$:
> $$a_0 x^5 - a_1 x \approx 0 \implies x \approx \left(\frac{a_1}{a_0}\right)^{1/4}.$$

2. **For small** $x$**:** The linear term $-a_1 x$ dominates over the other terms. Balancing $-a_1 x$ with $a_2$:
$$-a_1 x + a_2 = 0 \implies x \approx \frac{a_2}{a_1}.$$

3. **For intermediate** $x$**:** All terms contribute, and the roots must be found numerically or using perturbative methods.

---

**Reference Solution:**
Nondimensionalization We begin with the quintic polynomial:
$$a_0 x^5 - a_1 x + a_2 = 0.$$

To explore how the solution depends on the parameters $a_0$, $a_1$, and $a_2$, let us write:
$$x = \frac{a_2}{a_1} y,$$

and substitute this into the polynomial. This substitution yields:
$$a_0 \left(\frac{a_2}{a_1}\right)^5 y^5 - a_2 (y - 1) = 0.$$

Dividing through by $a_2$, the equation becomes:
$$\frac{a_0}{a_2} \left(\frac{a_2}{a_1}\right)^5 y^5 - y + 1 = 0.$$

Now define the parameter:
$$\epsilon = \frac{a_0 a_2^4}{a_1^5},$$

which simplifies the equation to:
$$\epsilon x^5 - x + 1 = 0.$$

We examine the roots of the equation:
$$\epsilon x^5 - x + 1 = 0$$

in two asymptotic limits: $\epsilon \to 0$ and $\epsilon \to \infty$. The aim is to match the expressions for the roots in the intermediate regime.

Consider the limit $\epsilon \to 0$. The equation involves three terms: $\epsilon x^5$, $-x$, and 1. For the equation to be satisfied, these terms must sum to zero. In the limit $\epsilon \to 0$, typically two of these terms dominate, forming a distinguished balance.

To discover approximate formulas for the roots, we analyze each possible balance and determine consistency.

Balancing $-x \approx 1$
The first possibility is to balance the terms $-x$ and 1. This implies, to a first approximation:
$$x \approx 1.$$

For this balance to be self-consistent, the neglected term $\epsilon x^5$ must satisfy:
$$\epsilon x^5 \ll 1.$$

Since $x \approx 1$, this condition reduces to $\epsilon \ll 1$, which is satisfied as $\epsilon \to 0$. Hence, this balance is consistent.

Balancing $x^5 \sim -1$
Consider the balance $\epsilon x^5 \sim -1$. This implies:
$$x \sim \left(-\frac{1}{\epsilon}\right)^{1/5}.$$

For this balance to be consistent, the neglected term $x \sim \epsilon^{-1/5}$ must be smaller than the terms we have kept. However, this is *not* the case. In the limit $\epsilon \to 0$, the neglected term grows unbounded, while the retained terms remain order unity. Therefore, this balance is not consistent.

Balancing $x^5 \sim x$

The only other possible balance is $\epsilon x^5 \sim x$. This balance leads to four roots:

$$x \sim \left( \frac{1}{\epsilon} \right)^{1/4}.$$

In this case, the size of the neglected term $1$ is much smaller than the size of the retained terms, which are of order $O(\epsilon^{-1/4})$. Thus, this balance is self-consistent.

Final Solution for the Quintic Roots

The quintic equation:
$$a_0 x^5 - a_1 x + a_2 = 0$$

has the following approximate roots depending on the coefficient regime, summarized below:

$$
\begin{aligned}
\text{Root 1:} \quad & x \approx 1, \\
\text{Roots 2–5:} \quad & x \approx \left( \frac{a_1^5}{a_0 a_2^4} \right)^{1/4} e^{i \frac{2\pi k}{4}}, \quad k = 0, 1, 2, 3.
\end{aligned}
$$

