# OpenReview forum: "HARDMath: A Benchmark Dataset for Challenging Problems in Applied Mathematics"
_ICLR.cc/2025/Conference — ICLR 2025 Poster_

### Official Review · Reviewer_m8U8 · 2024-10-31

**Soundness:** 3
**Presentation:** 2
**Contribution:** 3
**Rating:** 6
**Confidence:** 4

**Summary:**

This paper proposes a dataset of mathematical problems taken from a graduate level engineering course, and evaluates the ability of several LLMs on solving the problems in the dataset. The problems include topics related to polynomials, root finding, integrals, etc. Paper considers some base problems and modifies them randomly to generate a relatively large dataset. The dataset also includes some manually crafted problems. The accuracy of best LLM goes above 62% in solving the problems in the test set of this dataset. The paper goes on to analyze the failures of LLMs.

**Strengths:**

The topic is interesting and the dataset (at least, a good portion of it) can be useful for the community.

Paper is well written and the experiments are insightful.

**Weaknesses:**

In my view, the dataset could have consisted of more difficult problems. From what I read in the abstract and introduction, I was expecting a much harder set of problems to be included in the dataset. The relatively high accuracy of the GPT model on the dataset (above 60%) is also indicative that a considerable portion of the dataset consists of problems that are not as challenging.

Dataset generation method, described in Section 3.2, is still not completely clear to me, even after reading Appendix A. I think the generation procedures need to be explained in more detail such that one can reproduce the generation method by reading the paper and the appendices. Appendix A merely provides some examples of problems in the dataset which I was hoping to see at least one sample in page 1 or 2, rather than the appendix.

The topic of non-dimensionalizing of a polynomial does not strike me as interesting as some of the other topics. In the appendix A, paper provides exact formulas for non-dimensionalizing a polynomial - an exact formula does not need numerical methods as the paper suggests are the basis of all problems in the dataset. Quoting the claim from page 1: “most datasets focus on grade school- to high school-level mathematics problems whose solution methods only involve direct, ‘clean’ calculations. In contrast, HARDMATH targets applied mathematics problems that require approximate analytical solutions.”

When it comes to the so-called Word problems in the dataset, the paper crafts them manually, which is fine. But, it does not sit well with earlier claims of the paper such as “Rather than relying on the typical approach of collecting problems from textbooks, standardized tests, or competitions, as seen in most existing datasets, we developed algorithms to automatically generate problems and their step-by-step solutions.”

Randomly modifying the initial conditions of an ODE or coefficients of a polynomial can of course be done automatically. However, I think it is not fair to contrast such automation with the manual work that was the basis for some of the benchmarks in the literature.

**Questions:**

Please see weaknesses.

---

> ### Author Response · Authors · 2024-11-17
> **Response to Reviewer m8U8 (Part 1)**
>
> Thank you for your helpful suggestions!
>
> > Dataset generation method, described in Section 3.2, is still not completely clear to me, even after reading Appendix A. I think the generation procedures need to be explained in more detail such that one can reproduce the generation method by reading the paper and the appendices.
>
> Thank you for your suggestions! We have reworked **Section 3.2** to provide a clearer explanation of our dataset generation procedure. We also provide two new figures to demonstrate the process visually. **Fig. 2 (pg 4)**, is a flowchart detailing the problem generation process, which, together with the problem type descriptions in Section 3.3, can be used to reproduce the generation method. The second additional plot, Appendix **A.1, Fig. 5 (pg 13)** contains a visual demonstration of the validity checks performed on the problems in the evaluation set HARDMath-mini. Per your suggestion, we have moved a sample problem earlier with a full step-by-step solution to the **top of pg 4** of the main paper.
> However, we would also like to highlight that building the dataset generation code requires a certain level of expertise in applied mathematics and asymptotics. This underscores the particular value of ***HARDMath***: these problems are common in applied science and engineering, yet the methods to solve them are not widely known. To facilitate reproducibility, our codebase (attached, and to be made public after the double-blind review process) allows researchers to generate step-by-step solutions for any problem type in ***HARDMath***, or to reproduce the entire dataset and all results presented in the paper without requiring in-depth domain knowledge.
>
> > The dataset could have consisted of more difficult problems. The relatively high accuracy of the GPT model on the dataset (above 60%) is also indicative that a considerable portion of the dataset consists of problems that are not as challenging.
>
> Thank you for your feedback regarding the difficulty of the problems in the dataset. We'd like to address this concern from several angles:
>
> - We want to emphasize that our accuracy metric relies on partial credit. In other words, a 60% accuracy does not mean the model gets 60 problems correct out of 100 problems; instead, it is closer to getting a 60 out of 100 on an exam, where partial credit is generally assigned. Importantly, this means that a model may get all the final answers wrong, but still show “high accuracy,” because of our GPT-grading framework that focuses on specific aspects of the solution process. Figure 2 in our paper demonstrates that much of the credit given for *Roots, ODE and Integral* problems is partial—accuracy scores would decrease significantly without this partial credit method. Other benchmarks generally use absolute accuracy. For instance, on the MATH-500 benchmark (an updated version of the popular MATH benchmark), o1-mini is reported to achieve ***90%*** accuracy without any CoT prompting [1], but clearly performs much worse (even with partial credit) on our problems.
>
> - We would like to point out that evaluation accuracies and how they scale with CoT prompting are very uneven across individual problem types (**Table 2**, Appendix **A.4.1 Fig. 7** and **8, pg 28**). Take o1-mini as an example, relatively high overall accuracy (~60%) is dragged up by the easiest type (which you also mentioned in the review comment) of the problems - nondimensionalization.
>
> - Finally, we note that the relatively high evaluation accuracy of around 60% on our dataset was achieved specifically by OpenAI's o1-mini model, which is a recent model trained with a novel RL training pipeline that is specifically optimized for STEM reasoning (including mathematics and coding) [1]. This specialized training makes its performance on our dataset somewhat exceptional and not necessarily representative of other leading LLMs in general. Moreover, despite o1-mini's specialized training, it achieved ~60% accuracy only under 5-shot prompting conditions. In the 0-shot scenario, its accuracy was merely 29.8% (again, with partial credit), demonstrating that HARDMath-type problems are likely not present in o1-mini’s training data and that its innate reasoning capabilities do not yet cover this type of approximate reasoning.

---

> > ### Author Response · Authors · 2024-11-17
> > **Response to Reviewer m8U8 (Part 2)**
> >
> > > The topic of non-dimensionalizing a polynomial is not as interesting as some of the other topics—this can be solved exactly without numerical methods, which the paper suggests are the basis of all problems in the dataset.
> >
> > Thank you for your feedback. We agree that the nondimensionalization problems are fundamentally distinct from the other problem types in the dataset in that they can be solved exactly using algebra. In fact, it was our initial intention *not* to include nondimensionalization as a problem class in the dataset. As we mention, nondimensionalizing equations is an essential prerequisite for performing asymptotic analyses, doing perturbation theory, etc (see wikipedia.org/wiki/Nondimensionalization for details). When we began testing LLMs on the types of polynomials and ODEs in our dataset, we found they could not get past this preliminary nondimensionalization step. When we tested LLMs on nondimensionalization problems in isolation, we were surprised to find such a high failure rate. This is likely attributable to the fact that, as discussed in our paper, there are no existing benchmarks dedicated to applied math, so LLMs are unlikely to have seen such problems in their training corpus.
> >
> > Although methodologically, we agree that nondimensionalization problems are quite different than the rest of the dataset, we believe thematically, HARDMath is the right place to include such problems for two reasons: (1) Nondimensionalization is fundamental to solving problems in applied math, physics, fluid mechanics, and other applied sciences, so a thorough understanding of LLM performance on these problems is required when presenting a benchmark in this field. (2) If LLMs are finetuned or trained on our dataset, we thought it would be unuseful to have models that are capable of solving the more complex problem of performing asymptotic analyses on nondimensionalized equations, but that are unable to take a dimensionalized equation and nondimensionalize it to make these techniques applicable in real settings.
> >
> > Moreover, our results in **Table 2, Fig. 7** and **Fig. 8** demonstrate that all models except for o1-mini demonstrate relatively low accuracy on nondimensionalization problems under the 0-shot prompting case. Additionally, other models at few-shot prompting (e.g., GPT-3.5 1-shot, Llama3-8b, CodeLlama-13b) demonstrate a lower accuracy on nondimensionalization problems than on the human-perceived more difficult *ODE* or *Integral* types. This interesting and somewhat unexpected feature highlights the value of including nondimensionalization problems in HARDMath.
> >
> >
> > > Word problem dataset construction and modifying initial conditions of ODEs.
> >
> > We acknowledge that the manual crafting of the word problems falls outside of our otherwise fully-automated framework. In response to this, we have performed new experiments to demonstrate that this process of embedding realistic and relevant context into the problems in HARDMath can easily be automated and still generate diverse, high-quality problems. We provide an LLM (e.g., GPT-4o) with the problem and solution produced by our dataset generation framework and prompt it to output a real-world context embedding the problem and solution based on a specified domain seed (e.g. physics). We introduce another LLM as a verifier to check the plausibility of the word problem based on the given real-world context (e.g., in a physics setting, energy should always be non-negative), and produce a plausibility score. This procedure is detailed in the newly added **Section 3.5** in our revised PDF. Additionally, **Table 3 (Appendix A.2.7, pg 23)** in our revised paper documents the prompts used for this context embedding and verification, and Appendices **A.1.8 (pg 24)** and **A.2.9 (pg 25)** confirm that these generated problem contexts are high-quality.
> > Importantly, we also wish to clarify that while modifying problem coefficients or initial conditions is relatively simple, automatically generating the corresponding step-by-step solutions and ensuring their correctness using self-verification tools is highly non-trivial. Importantly, our solutions are not mere interpolations of a numerical solver but are closed-form analytical approximations that require both “pen-and-paper” and numerical calculations. This automatic solution generation is our main contribution. To clarify this, we have added more details about this solution generation process in **Section 3.2 (Fig. 2, pg4)** and described the semi-automatic validity check procedure in **Section 3.2** and **Fig. 5 (pg 13)**.
> >
> >
> > [1] OpenAI. (2024, September 12). OpenAI o1-mini: Advancing cost-efficient reasoning. OpenAI. Retrieved from https://openai.com/index/openai-o1-mini-advancing-cost-efficient-reasoning/

---

> > > ### Comment · Reviewer_m8U8 · 2024-11-18
> > >
> > > I thank the authors for their detailed and clear response, and the additional experiments. These improvements address most of my questions/concerns, and I think the proposed dataset will be useful for the community. I raise my score from 6 to 7.
> > >
> > > I understand the motivation behind the nondimensionalization problems in the dataset, and it would be useful to include the intermediate results (that led the authors to include those problems in the dataset) in some appendix.
> > >
> > > I would like to point out again that some of the claims in the paper about the previous works in the literature being manual and the work in this paper being fully automated does not seem accurate to me. In my view, the authors' method involves considerable human crafting and judgement which is a good thing. And I think the paper does not need to distance itself from the previous work by assigning those labels.

---

> > > > ### Author Response · Authors · 2024-11-19
> > > > **Thanks to m8U8 and updated writing**
> > > >
> > > > Thank you so much for your consideration and for increasing your score!
> > > >
> > > > > **It would be useful to include the intermediate results (that led the authors to include those problems in the dataset) in some appendix**
> > > >
> > > > * We appreciate this insight and have added a new example to the paper to show how LLMs could not correctly answer this preliminary nondimensionalization step in the case of *polynomial* and *ODE* examples. In particular, we show GPT-4’s answer when prompted to find the roots of a *dimensionalized* polynomial against the ground-truth solution in **Appendix A.5 (pg 32)**. This example highlights that the model doesn’t even consider the natural initial nondimensionalization step before starting its analysis of different regimes and approximation processes. Then, the LLM solution entirely fails to capture the correct roots, and does not even recognize that it is missing the requirement for $n=5$ roots in a $5^{th}$-order polynomial. Without the nondimensionalization, the number of possible coefficient combinations is much larger, making it harder to find correct solutions.
> > > >
> > > > We also understand your point regarding the distinction between manual and automatic data generation—you are correct that our approach is not fully automatic. Rather, it involves significant human input in selecting problems, creating solution frameworks, and validating model responses, which are all important when curating a reliable benchmark. To better reflect this, our revised manuscript emphasizes our hybrid approach, which combines automated problem/solution generation (enabling scalable datasets), and automated problem grading (giving us a scalable evaluation method), while still acknowledging the relevant and important human design and feedback steps in our pipeline. We agree that the paper should better reflect the value of human crafting in the process and have adjusted the language accordingly.

---

### Official Review · Reviewer_2kK3 · 2024-11-02

**Soundness:** 3
**Presentation:** 3
**Contribution:** 3
**Rating:** 6
**Confidence:** 4

**Summary:**

The author introduces a challenging math dataset focused on asymptotic reasoning, highlighting the use of mathematical approximations to address complex problems that commonly arise in real-world scenarios.

**Strengths:**

1- Introducing a dataset that requires approximate analytical solutions is innovative, addressing a gap in current benchmarks. This approach aligns with many real-world problems and reflects how scientists typically tackle them.

2- The dataset is larger than similar graduate-level math datasets. It can be utilized to develop novel prompting techniques or for fine-tuning models.

**Weaknesses:**

The authors suggest that their dataset could be utilized for fine-tuning LLMs to enhance their capabilities. However, it would be valuable to see empirical results demonstrating this improvement.

The authors explore one-shot and five-shot CoT prompting, noting a substantial performance increase across most models. It would be worthwhile to investigate more complex CoT prompting techniques

**Questions:**

Refer to the weakness of paper section.

---

> ### Author Response · Authors · 2024-11-17
> **Response to Reviewer 2kK3**
>
> Thank you for your thoughtful review! Our responses are below.
>
> > **Q1: The authors suggest that their dataset could be utilized for fine-tuning LLMs to enhance their capabilities. However, it would be valuable to see empirical results demonstrating this improvement.**
>
> This is a valuable suggestion! However, while we agree that domain-specific fine-tuning with our dataset is a promising future direction, we believe it is beyond the scope of this paper for several reasons:
>
> 1. Our primary objective here is to introduce the HARDMath dataset, establish baseline performance benchmarks for existing LLMs, and highlight the limitations of LLM reasoning in this important domain of advanced mathematics. This approach aligns with standard practice in the field, where new datasets are often introduced independently of fine-tuning experiments [1-3].
>
> 2. In the paper, we note that the HARDMath dataset is available for further model development, including fine-tuning as well as other techniques commonly used in mathematical domains, such as self-verification [4] and tool usage [5]. We believe a thorough investigation of diverse techniques that could improve LLMs’ performance on these domain-specific advanced mathematical problems, including fine-tuning, would be worthy of a separate paper. Notably, these improvements depend on the robust benchmarking framework introduced in this paper.
> Although we leave fine-tuning to future work, we believe that the dataset will be useful for fine-tuning because:
>
> - we provide code for generating the dataset and the self-verification tool (see newly modified **Section 3.2** and **Figs. 2, pg 4 and 5, pg 13** for more details), enabling researchers to produce data as needed. Compared to datasets created by scraping textbooks, our approach avoids limitations in data volume and answer quality, enhancing fine-tuning potential.
>
> - Our solution sets include not only final answers but also detailed, step-by-step procedures. For difficult mathematical questions, OpenAI's work [6] demonstrates that process supervision—rewarding correct steps rather than just outcomes—improves alignment and performance in mathematical reasoning tasks. A dataset like ours that contains stepwise reasoning could serve as a basis for developing such a process supervision reward model.
>
> > **Q2: The authors explore one-shot and five-shot CoT prompting, noting a substantial performance increase across most models. It would be worthwhile to investigate more complex CoT prompting techniques**
>
> Thank you for your appreciation of our results in comparing different shots of CoT prompting. We interpret the meaning of “complex CoT prompting techniques” to be either CoT prompting with more examples, or more recent techniques that come after CoT.
>
> For the former, we actually include results on CoT prompting to up to 10 examples in Appendix **A.4 Fig. 8 (pg 28)** for all models except for o1-mini (due to cost constraints). Based on these results, we actually do not see a substantial performance increase across most models, and believe that the performance increase is both model and problem-dependent. For example, performance on ODEs across all models increases minimally beyond 1-shot CoT prompting.
>
> To the latter point, we agree that it would be interesting to see how more sophisticated prompting techniques could improve performance on our challenging domain-specific dataset. However, as mentioned previously, we think this moves away from the main focus of this paper, which is to provide the dataset for the community, establish baseline performance benchmarks for existing LLMs, and provide insight into the failure modes of LLM reasoning on this essential area of advanced mathematics. In this setting, we intentionally selected the most popular prompting techniques for our evaluations because they are consistent with other benchmarking studies—for example, the Llama 3, GPT, and o1 series models all use standard CoT prompting [7-9].
>
>
> [1] Lu et al. (2023), MathVista: Evaluating Math Reasoning in Visual Contexts
>
> [2] Sawada et al. (2024), ARB: Advanced Reasoning Benchmark for Large Language Models
>
> [3] Frieder et al. (2023), Mathematical Capabilities of ChatGPT
>
> [4] Weng and Zhu et al. (2022), Large Language Models are reasoners with Self-Verification
>
> [5] Schick et al. (2023), Toolformer: Language Models Can Teach Themselves to Use Tools
>
> [6] Lightman et al. (2023), Let’s Verify Step by Step
>
> [7] OpenAI (2024). GPT-4 Technical Report
>
> [8] OpenAI. (2023). Simple Evals. GitHub repository. Retrieved from https://github.com/openai/simple-evals
>
> [9] Meta AI. (2023). LLaMA 3. Retrieved from https://ai.meta.com/blog/meta-llama-3/

---

> > ### Author Response · Authors · 2024-12-02
> > **Follow-up to 2kK3**
> >
> > Dear reviewer, we wanted to ask if there’s anything else we can do to address your comments. You had mentioned that it would be worthwhile to investigate more complex CoT prompting techniques—we note that our paper includes up to 10-shot CoT experiments for all models excluding o1-mini. We have also clarified many other aspects of the paper in the updated document, including the data generation process, to explain how it can be used for fine-tuning models.

---

### Official Review · Reviewer_9Y89 · 2024-11-04

**Soundness:** 3
**Presentation:** 3
**Contribution:** 3
**Rating:** 8
**Confidence:** 1

**Summary:**

This paper proposes a new math benchmark for evaluation of LLMs.

**Strengths:**

This paper makes an important contribution to the LLM community. The LLM community needs a harder math benchmark with the growing capability of the models. I think this benchmark will be useful.

**Weaknesses:**

I don't see obvious weaknesses.

**Questions:**

Have the authors evaluated larger open-source models such as Llama 3.1 405B?

---

> ### Author Response · Authors · 2024-11-17
> **Response to Reviewer 9Y89**
>
> Thank you for your valuable feedback and your appreciation of our work! We are glad to read that you share our view on the need of harder benchmarkers to differentiate the ever-evolving frontier LLMs. We have provided our responses below.
>
> **Q1: Have the authors evaluated larger open-source models such as Llama 3.1 405B?**
>
> We have not run evaluations on larger open-source models like Llama 3.1 405B due to limited access to computing resources. According to Meta’s official hardware guide [1], deploying a 405B model even at 8-bit mode needs 8 x NVIDIA A100 or H100 GPUs plus multi-core high performance CPU with 256GB+ RAM. Such a configuration is challenging to implement stably in our setting. However, we would like to point out that based on Meta’s Llama 3.1 405B benchmarking results [2], this model has comparable performance on mathematical benchmarks GSM8K and MATH to the closed-source model GPT-4. Therefore, we extrapolate that our evaluation results on GPT-4 could be representative of open-source models of the same tier, such as Llama 3.1 405B. Additionally, Llama 3.1 405B is also similar to GPT-4 in terms of model parameter size. We encourage the reviewer to take a look at Appendix **A.4 Fig. 8 (pg 28)**, where we summarize the scaling of evaluation accuracy given shot number, problem type and model parameter size.
>
> [1] "Hardware Requirements for Llama3-1 Model." Llama3-1, 2024. Available at: https://llama3-1.com/405b/hardware-requirements/. Accessed November 12, 2024
>
> [2] Introducing Llama 3.1." Meta AI Blog, 2024. Available at: https://ai.meta.com/blog/meta-llama-3-1/. Accessed November 12, 2024

---

### Official Review · Reviewer_dUZT · 2024-11-04

**Soundness:** 3
**Presentation:** 3
**Contribution:** 2
**Rating:** 6
**Confidence:** 4

**Summary:**

This work presents a new math dataset that targets on applied mathematics and requires graduate level knowledges. The problems in this dataset demand a combination of mathematical reasoning, computational tools, and subjective judgment. For example, it may need checks for self-consistency and the use of numerical methods.

**Strengths:**

1. This paper presents a method to generate graduate level math problems (in some pre-defined areas) and corresponding solutions.

**Weaknesses:**

1. The type of the math problems in this dataset is limited. First, only three main subjectives are discussed, ODEs, integrals and polynomial. Second, most of the problems are related to calculations, and it seems none of them is proof related. For such problems, the existed numerical computational software should be able to solve most of them. In summary, the diversity of this dataset is limited.

2. Since the dataset can be generated with python tools, it is possible to generate more examples for training. I didn't see any experiment about training on such dataset.

**Questions:**

1. The description in Section 3.2 is not clear. For example, the paper presents that `sympy` and `scipy` are used to implement the mathematical procedures required for obtaining approximate, analytical solutions. In Appendix A, it only includes some examples with questions and solutions, and I did not see the role of `sympy` and `scipy`.

2. As mentioned in Weakness 2, besides constructing the benchmark, I think this work has the potential to be used for constructing large scale training dataset, teaching the LLMs to solve such hard math problems with the suggested thoughts in CoT, or generated `sympy` code in PoT. What do the authors think about this direction?

3. As my concern of diversity in the Weakness 1, I also have the same concern on the solution ideas. It seems that many problems share the same solution idea by just randomly changing the numbers in the problem. This may be verified by the large performance increase from 0-shot to 5-shot. As the 5 demonstration examples may have already provide the correct solution path, the LLMs only need to change some numbers. Can the author introduce how many solution strategies are used for each type in your dataset?

---

> ### Author Response · Authors · 2024-11-17
> **Response to Reviewer dUZT (Part 1)**
>
> Thank you for your thoughtful review! We have added several new figures and respond to your suggestions below.
>
> **Questions:**
>
> > **The description in Section 3.2 is not clear.**
>
> We appreciate the reviewer’s feedback and acknowledge the connections between SymPy/SciPy and their use could be clearer. We have therefore added a new flowchart in Fig. 2 (pg 4) of the revised paper to explain the data generation process visually, including references to the use of SymPy and Scipy.
>
> SymPy derives symbolic, analytical forms of approximate solutions. It is crucial for
> - simplifying equations
> - performing series expansions for asymptotic approximations
> - solving symbolic systems to extract dominant balances and higher-order corrections.
>
> SciPy performs numerical validation; providing “ground-truths” for validating accuracy of analytical solutions. Specifically, it
> - approximates integrals and ODEs numerically
> - computes numerical roots of polynomials for comparison
> - allows for relative error calculations between numerical and analytical solutions to ensure we include only correct solutions.
>
> Examples in Appendix A are meant to present problems and their solutions in a human-readable form, but every solution relies on the computational tools above. SymPy handles symbolic derivations, while SciPy verifies their correctness against numerical results. The codebase included as supplemental material in our submission includes scripts explaining problem and solution generation.
>
> > **This work has the potential to be used for constructing large scale training dataset**
>
> We think this is a great idea, but since our focus here is on benchmarking models, highlighting limitations in frontier LLMs, and providing a dataset for the research community to use, we do not include experiments along this direction in this paper.
>
> In the paper, we echoed your point that the HARDMath dataset offers a good platform for developing and testing advanced techniques that could improve LLMs' domain-specific performance on these challenging math problems. We believe a thorough investigation of diverse techniques, e.g. the prompting techniques you mentioned (e.g. PoT, ToT [1-2]), fine-tuning, or tool usage [3], would be worthy of a separate paper. Notably, these improvements would depend on the robust benchmarking framework introduced in this paper.
>
> > **Can the author introduce how many solution strategies are used for each type in your dataset?**
>
> We appreciate the reviewer’s concern regarding solution diversity but suggest this point does not fully recognize the dataset's structure and the nature of the applied mathematics problems. Even slight changes to the numbers can drastically change the solution, as they may change the dominant balance, necessitate higher-order corrections, or render a previously valid solution regime invalid. There are at least 3–5 distinct solution strategies per problem type, which are outlined in detail in the Appendix.
>
> Each problem type involves distinct,non-trivial multi-step strategies. Some techniques we use include scaling transformations, asymptotic expansions, dominant balance, Taylor expansions, and numerical computations to guide solution strategies. Each problem type involves different combinations of these techniques and could depend on the problem’s specific regime (e.g., small vs. large parameter values). The newly added Box 1 (pg 4) is an example of how dominant balance and physical approximation techniques are combined in the solution.
>
> Additionally, the performance improvement seen with few-shot CoT prompting is expected and desirable, as it shows the model can generalize examples. However, we note that this improvement is limited by the inherent difficulty and diversity of our dataset, as evidenced by even the best-performing models achieving far from perfect accuracy, and the fact that performance improvements are relatively small. Performance on more complex problems like ODEs also increases minimally across models beyond 1-shot CoT prompting (Section 4.3, Appendix A.4 Fig. 8, pg 28).
>
> **Finally, we want to emphasize that our accuracy metric relies on partial credit.** In other words, a 30% accuracy does not mean the model gets 30 problems correct out of 100; instead, it can more aptly be interpreted as, say, getting 30/100 on an exam, where partial credit is generally assigned. Notably, this means that a model might get all the final answers wrong but still show “high accuracy,” because our GPT-grading framework focuses on the solution process and answer rather than only the final answer. Fig. 3 (pg 9) in our paper demonstrates that much of the credit given for Roots, ODE and Integral problems is partial—accuracy scores would decrease significantly without this partial credit method. The accuracy improvement with CoT demonstrates that the model is learning the problem-solving strategy, not just changing numbers in the solution.

---

> ### Author Response · Authors · 2024-11-17
> **Response to Reviewer dUZT (Part 2)**
>
> **Weaknesses:**
>
> 1. Diversity of the dataset
>
> The goal of the HARDMath dataset is to focus on a subset of challenging problems in applied mathematics, specifically those that require asymptotics and approximations. We maintain that nonlinear ODEs, complex integrals, and polynomial root-finding are central to applied science and engineering fields, and comprise a large portion of these fields. For instance, two-thirds of the widely-used graduate textbook, “Advanced Mathematical Methods for Scientists and Engineers,” by Bender and Orszag [1] are dedicated to these methods, and directly states, ***“The mathematical methods discussed in this book… are the most useful and powerful methods for finding approximate solutions to equations, but they are difficult to justify rigorously”*** [1]. As such, these methods are fundamental for applied mathematics and do not involve proof-based techniques, and our focus is on benchmarking LLM reasoning capabilities specific to applied science settings. Numerous existing datasets such as GHOSTS [2] already focus on proofs and theoretical mathematics, but few cover important graduate-level applied mathematics skills, and none to the scale of HARDMath.
>
> On the topic of numerical vs. analytical solutions, we would like to emphasize two points. First, on many of the types of equations we have included, numerical solutions can be highly inaccurate, and detecting these inaccuracies is often not straightforward. The second point, and the reason why the use of asymptotics and perturbation theory underlies almost all theoretical work in the sciences, is because a numerical solution by itself offers very limited value: numerical outputs from a solver alone provide neither analytical insights about the equation nor intuition about the system at hand. Asymptotic analyses are so fundamental in physics because physicists do not merely want to know the numerical solution to an equation at some point in the domain, but rather to formulate equations that accurately capture scaling behavior, enable them to identify leading order contributions (e.g, what physical effects are controlling the behavior), etc. Numerical solutions do not enable these types of studies. Reasoning based on asymptotics truly forms the basis for much of the sciences, and we feel strongly that equipping LLMs with the ability to interact with numerical solvers is not a replacement for this type of reasoning.
>
> In HARDMath, our analytical solutions are validated against numerical results, and this combination allows for a more robust understanding of the problem and solution—something numerical solvers alone cannot achieve. We do believe a benchmark for LLM numerical solver use would be valuable for the community, but think it would target a fairly different direction (similar to tool usage and agents) rather than mathematical reasoning.
>
> 2. Training on the dataset
>
> It is certainly possible to generate more examples using our framework and fine-tune an LLM on our dataset. However, we believe that fine-tuning is beyond the scope of this paper. Our primary objective here is to introduce the HARDMath dataset, establish baseline performance benchmarks for existing LLMs, and highlight the limitations of LLM reasoning in this important domain of advanced mathematics. This approach aligns with most other papers in the field that introduce datasets separately from fine-tuning experiments [3–5]. However, we mention in Section 5 that fine-tuning on an expanded dataset is our immediate next step and believe our framework will be effective for this since we provide code for data generation.
>
> [1] Yao et al. (2023), Program of Thoughts Prompting: Disentangling Computation from Reasoning for Numerical Reasoning Tasks
>
> [2] Yao et al. (2024), Tree of Thoughts: Deliberate Problem Solving with Large Language Models
>
> [3] Schick et al. (2023), Toolformer: Language Models Can Teach Themselves to Use Tools
>
> [4] Bender and Orszag (1999), Advanced Mathematical Methods for Scientists and Engineers I
>
> [5] Frieder et al. (2023), Mathematical Capabilities of ChatGPT
>
> [6] Lu et al. (2023), MathVista: Evaluating Math Reasoning in Visual Contexts
>
> [7] Sawada et al. (2024), ARB: Advanced Reasoning Benchmark for Large Language Models
>
> [8] Frieder et al. (2023), Mathematical Capabilities of ChatGPT

---

> > ### Comment · Reviewer_dUZT · 2024-11-27
> >
> > Thanks for the further clarification. I increased my score.

---

> > > ### Author Response · Authors · 2024-11-27
> > >
> > > Thank you for your appreciation of our response and work.

---

### Author Response · Authors · 2024-11-23
**Overall response to reviewer**

We thank the reviewers for their time and comments! We are glad that the reviewers acknowledge the need for a harder math benchmark that addresses the growing capabilities of LLMs, and the dataset's potential use for improving future LLMs. The comments that our experiments were well-designed and well-presented are especially appreciated!

We have updated the text of our paper in response to the reviewers’ suggestions and added several new figures/sections. Here is a summary of our main changes.

1. To integrate dUZT’s suggestions regarding the use of SymPy/SciPy and the generation process, we added more text details in **Section 3.2** and a new flowchart in **Fig. 2 (pg 4)** of the revised paper to explain the data generation process visually. We also incorporated **Box 1 (pg 4)** to demonstrate diverse solving strategies used in an example solution.

2. To address the question that evaluations for some model/conditions reaches relatively high accuracy (~60%), we emphasize that our accuracy metric relies on partial credit, which means that the ~60% accuracy rate is a result of our GPT-grading framework that focuses on the solution process and answer rather than only the final answer. **Fig. 3 (pg 9) in our paper demonstrates that much of the credit given for Roots, ODE, and Integral problems is partial—accuracy scores would decrease significantly without this partial credit method.**

3. On the topic of the diversity and relevance of our dataset, we note that the fields of applied mathematics and mathematical physics **rely** on these problem classes. For example, when referring to asymptotic and approximation methods, a widely-known textbook writes that they “are the most useful and powerful methods for finding approximate solutions to equations, but they are difficult to justify rigorously” [1]. Few datasets cover important graduate-level applied mathematics skills, and none to the scale of HARDMath.

We also better motivate the inclusion of nondimensionalization, since it is a fundamental technique used in our other approximation methods, which current models fail at (**Appendix A.5, pg 32**).

4. we add a new method for automatically constructing a word problem dataset using LLMs like GPT-4o (please see the new **Section 3.5, pg 6, and Table 3, A.2.7, pg 23**). In this framework, we provide the base mathematical problem and a subject domain, are given a word problem by the model, and then evaluate the topic diversity and “plausibility” of the problem using another LLM (A.2.8, A.2.9, pgs 24-25). We only select problems with a plausibility score above a minimum and show that many realistic and diverse word problems can easily be generated.

We again thank the reviewers and the AC for their time.

---

> ### Author Response · Authors · 2024-11-25
>
> We thank again the reviewers for their time and comments! Since the discussion period is ending soon, we’d like to kindly ask if there is anything else we can provide the reviewers with to address any remaining questions!

---

### Meta-Review · Area_Chair_1eXv · 2024-12-21

**Metareview:**

The authors introduce HARDMath, a new benchmark dataset for evaluating LLMs' capabilities for solving graduate-level applied mathematics problems. Types of problems included in this dataset include polynomial non-dimensionalization and rootfinding, nonlinear ODEs, traditional and Laplace integrals, and "word problems" (essentially, the previous types of problems contextualized in a quasi-realistic setting).

The paper is well written. The benchmark seems reasonable and thoughtfully constructed, and the authors do a reasonable job profiling a handful of existing LLMs. The fact that the questions are algorithmically generated is a plus, and may be helpful for mitigating memorization. One concern is that benchmarks such as this one might quickly become obsolete, especially with the current pace of progress in reasoning-focused LLMs such as the o-series from OpenAI. Looking ahead I'd encourage the authors to think about how to ensure long-term usage of this benchmark by the community.

**Additional Comments On Reviewer Discussion:**

Most of the initial review sentiment was positive. There were some questions raised about possibly using a part of this dataset for fine-tuning existing LLMs, but I suppose that's a topic for follow-up work. The authors responded well to most questions.

---

### Decision · Program_Chairs · 2025-01-22

Accept (Poster)